# Subcellular pathways through VGluT3-expressing mouse amacrine cells provide locally tuned object-motion-selective signals in the retina

Karl Friedrichsen [1,2,3,4], Jen-Chun Hsiang [1,2,3,4], Chin-I Lin [1,2,3,4], Liam McCoy[1,2,3], Katia Valkova[1,2,3], Daniel Kerschensteiner [1,2,3] ✉ & Josh L. Morgan [1,2,3] ✉

VGluT3-expressing mouse retinal amacrine cells (VG3s) respond to small-object motion and connect to multiple types of bipolar cells (inputs) and retinal ganglion cells (RGCs, outputs). Because these input and output connections are intermixed on the same dendrites, making sense of VG3 circuitry requires comparing the distribution of synapses across their arbors to the subcellular flow of signals. Here, we combine subcellular calcium imaging and electron microscopic connectomic reconstruction to analyze how VG3s integrate and transmit visual information. VG3s receive inputs from all nearby bipolar cell types but exhibit a strong preference for the fast type 3a bipolar cells. By comparing input distributions to VG3 dendrite responses, we show that VG3 dendrites have a short functional length constant that likely depends on inhibitory shunting. This model predicts that RGCs that extend dendrites into the middle layers of the inner plexiform encounter VG3 dendrites whose responses vary according to the local bipolar cell response type.

In a simplified neuron model, all dendritic inputs sum to trigger a unified output signal. However, the spatial distribution of input synapses can influence dendritic computations in complicated ways (reviewed in ref. 1). Building realistic models of neurons and their connectivity requires understanding how the spatial distribution of synapses relates to the subcellular integration and transmission of signals. Neurons, where inputs and output synapses occur on the same dendrites, provide an extreme test case for such analyses.

VG3 amacrine cells, like most retinal amacrine cells, lack an axon and do not fire action potentials[2,3]. Instead, they form input and output synapses on their dendrites. Unlike other amacrine cells, VG3s release both glutamate and glycine[2,4]. VG3s transform the motion-sensitive light responses of transient bipolar cells into a small-object-motion selective signal transmitted to RGCs and other amacrine cells. VG3s

perform this function by integrating excitatory input from ON and OFF transient bipolar cells with inhibitory input from global-motion sensing amacrine cells[3,5-7]. VG3s then drive small object motion responses in multiple ON, OFF, and ON/OFF RGC types (glutamatergic output[5,6,8]) and suppress activity in Suppressed-by-Contrast RGCs (glycinergic output[9,10]). VG3s respond robustly to looming stimuli, small dark objects that increase in size, and are required for the innate fear responses of mice to these stimuli[5].

Despite VG3s being small field (~100 μm) amacrine cells, different parts of their arbors exhibit distinct light responses. Dendrites differ in their contrast preferences (ON vs OFF, or polarity), receptive field size and looming sensitivity depending on their depth within the ON and OFF sublamina of the inner plexiform layer (IPL)[7,8]. Different dendrites of a single VG3 can also represent different regions of visual space, and

[1]Department of Ophthalmology and Visual Sciences, Washington University in St. Louis, St. Louis, MO, USA. [2]Department of Neuroscience, Washington University in St. Louis, St. Louis, MO, USA. [3]Department of Biomedical Engineering, Washington University in St. Louis, St. Louis, MO, USA. [4]Graduate Program in Neuroscience, Washington University in St. Louis, St. Louis, USA. ✉e-mail: kerschensteinerd@wustl.edu; jlmorgan@wustl.edu

thus the VG3 plexus encodes stimulus positions with subcellular precision[7,8].

The mixing of input and output synapses, compartmentalization of signals, and diverse input and output types all argue that VG3 signal processing should be understood as a system of subcellular pathways. Mapping these pathways requires tools to characterize structure-function relationships with subcellular resolution. Previous studies of dendritic processing have used subcellular calcium imaging to understand the extent of signal spread in amacrine cell arbors[11–13] while serial section electron microscopy has been used to map the fine-scale distribution of synaptic connections across amacrine cell arbors[14–17]. Here we combine the two techniques in the same tissue. We first characterize the polarity of the ON/OFF light responses in the dendrites of a mouse VG3 plexus with calcium imaging. We then use 3D electron microscopy (3DEM) to reconstruct the synaptic connectivity of the same functionally characterized VG3 plexus. By combining these modalities, we generated a subcellular wiring diagram of signal flow through a plexus of mixed input/output dendrites.

## Results

### Calcium imaging of VG3 dendrites

We used 2-photon imaging of the calcium indicator GCaMP6f to record light responses in individual VG3 dendrites. The retina of a P68 (postnatal day) VG3-Cre/Ai148 transgenic mouse (GCaMP6f expressed in VG3s) was removed and imaged live as described previously[8]. Calcium responses were recorded in two nearby regions (Fig. 1a, b). Each region was sampled at six depths covering the ON and OFF strata of the central IPL. Flashing bar stimuli (width: 50 μm, height: 60 μm) were presented at 1.5 s intervals. This stimulus size should equally drive ON and OFF bipolar cells innervating VG3s (Supplementary Fig. 1). We defined response polarity as the ON response minus the OFF response divided by their sum. We determined the precise depth of these recordings by identifying dendrite positions within the subsequent EM dataset (Fig. 1c, see below). The response polarities of the VG3 dendrites in this sample were consistent with previous studies[8]. VG3 dendrites in the OFF sublamina responded predominantly to OFF stimuli (i.e., negative polarity), whereas dendrites in the ON sublamina had more mixed responses (i.e., neutral polarity, Fig. 1c, d).

### 3DEM reconstruction of functionally characterized VG3s

After functional characterization, the retina was fixed, optically mapped at multiple resolutions, and then fixed and stained for EM reconstruction. We cut the tissue block into 1000 40 nm thick sections and mapped them with low-resolution EM. By comparing optical maps to low-resolution EM (Fig. 2a–c, see ref. 18 for details), we targeted the collection of two additional image volumes (Fig. 2d, 100 μm × 100 μm × 40 μm at 4 nm resolution and 400 μm × 400 μm × 40 μm at 20 nm resolution) to the functionally characterized VG3s. We manually traced the arbors of VG3s using VAST[19](Fig. 2e, f, Supplementary Table 1, Supplementary Movie 1) and identified 1835 synaptic inputs and outputs (Fig. 2g, Table 1). All cell reconstructions are considered partial because tracing was limited by image ambiguities and the edges of the imaged volume. However, a comparison of the arbor sizes of the most complete VG3 reconstructions (convex hull area = 3076, 3830, 4324, 4562 μm², Table 1) with the arbor sizes observed in previously published optical images of VG3 cells ($n = 66$ from 2410 to 6739, $4723 \pm 120$[20], $7662 \pm 211$ μm², $n = 39$[5],) indicates that most of the arbors were reconstructed for some VG3s.

For most of our results, we present averages and standard errors with an $N$ equal to the six most reconstructed VG3s. These samples are not independent as they are all taken from the same region of the same mouse retina and are often connected to the same synaptic partners. The standard errors presented, therefore, reflect our confidence in estimating parameters within a limited set of cells.

The reconstructed VG3 plexus spanned from 16% to 60% of IPL depth (95% length contained), consistent with previous reports of VG3s and the boundaries of the ChAT bands[21]. One amacrine cell (VG3 #5, Table 1, Supplementary Table 2), identified as a VG3 in optical imaging by its expression of the cre-driven calcium reporter, had arbor morphology strikingly different from other VG3s (Supplementary Fig. 2). Most of its dendrites branched relatively little, forming an asymmetrical, sparse arbor with incomplete coverage of its dendritic field. Though its arborization differed from other VG3s, its synaptic connectivity (see below) was closely matched.

### Differential distribution of excitatory and inhibitory inputs to VG3s

We identified 1777 input synapses and 1279 output synapses across six VG3 cells (Table 1). The input and output synapses were mixed throughout the VG3 arbor such that there was no distinct input or output region (Fig. 3a, Supplementary Fig. 2). Ribbon synapses from bipolar cells comprised $26.1 \pm 1.9\%$ (SE for $n = 6$) of the input synapses, while conventional synaptic inputs from amacrine cells comprised $73.9 \pm 1.9\%$. This excitation to inhibition (E/I) balance, $0.35 \pm 0.035$, is much lower than what is typically reported for amacrine and RGCs[22]. To identify the cell types postsynaptic to VG3s, we traced postsynaptic dendrites until we encountered an output synapse (amacrine cell) or passed several branch points with no synaptic outputs (RGC). We did not identify any VG3 synapses innervating bipolar cells or other VG3s. Targets were split roughly equally between other amacrine cells ($57.3 \pm 3.0\%$) and RGCs ($42.8 \pm 2.9\%$).

The spatial distribution of excitatory and inhibitory synapses differed in a way that could impact signal integration in VG3s. Bipolar inputs were concentrated in two strata corresponding to the axon terminals of OFF and ON bipolar cells, separated by a gap (Fig. 3b–d). The density of amacrine cell inputs was highest in the gap between the ON and OFF bipolar cell input strata and on the primary dendrites of VG3s. Consistent with this distribution, the calcium response amplitudes to visual stimuli were lower in the middle of VG3 arbors (Supplementary Fig. 3). The distribution of inhibitory inputs appears well-positioned to dynamically regulate signal spread between the ON and OFF dendrites of VG3s and between different primary dendrites of the same VG3. Further, the lower density of bipolar cell inputs on these potential crossover regions could contribute to the segregation of ON and OFF signals in the VG3 arbor.

A previous analysis of synapse distribution on AII amacrine cells found that outputs tended to be located at the tips of dendrites[23]. In VG3s, we observed increased bipolar cell input density and decreased amacrine cell input near dendrite tips (Fig. 3e, f). To test if bipolar cells had inhibition-free paths to RGC outputs near dendrite tips, we checked for different combinations of bipolar cell inputs, amacrine cell inputs, and output synapses to RGCs. We found that the frequency of bipolar cell and RGC synapses occurring together in the region of low amacrine density was too low (0.61% of 1851 tips) to be considered a significant contributor to VG3 transmission.

### VG3s are innervated by all transient bipolar cells but preferentially synapse with type 3a

We attempted to reconstruct and identify all bipolar cells forming ribbon synapses onto the reconstructed VG3s. We were able to identify the types of 101 bipolar cells forming 428 of the 477 (89.7%) ribbon synapses. We identified bipolar cell types by comparing the morphology, stratification depth, branch angle, and branch depth to the bipolar cell-cell types in the Eyewire Museum[24]. Distinctions between most bipolar cell types could easily be made based on stratification depth (Fig. 4a). However, distinctions within the transient OFF group (3a, 3b, and 4) and type 5 ON bipolar cell group (5t, 5i, and 5o) were more difficult, and we used a combination of arbor morphology and tiling rules to assign cell types (Fig. 4b, c). Bipolar cells of the same type

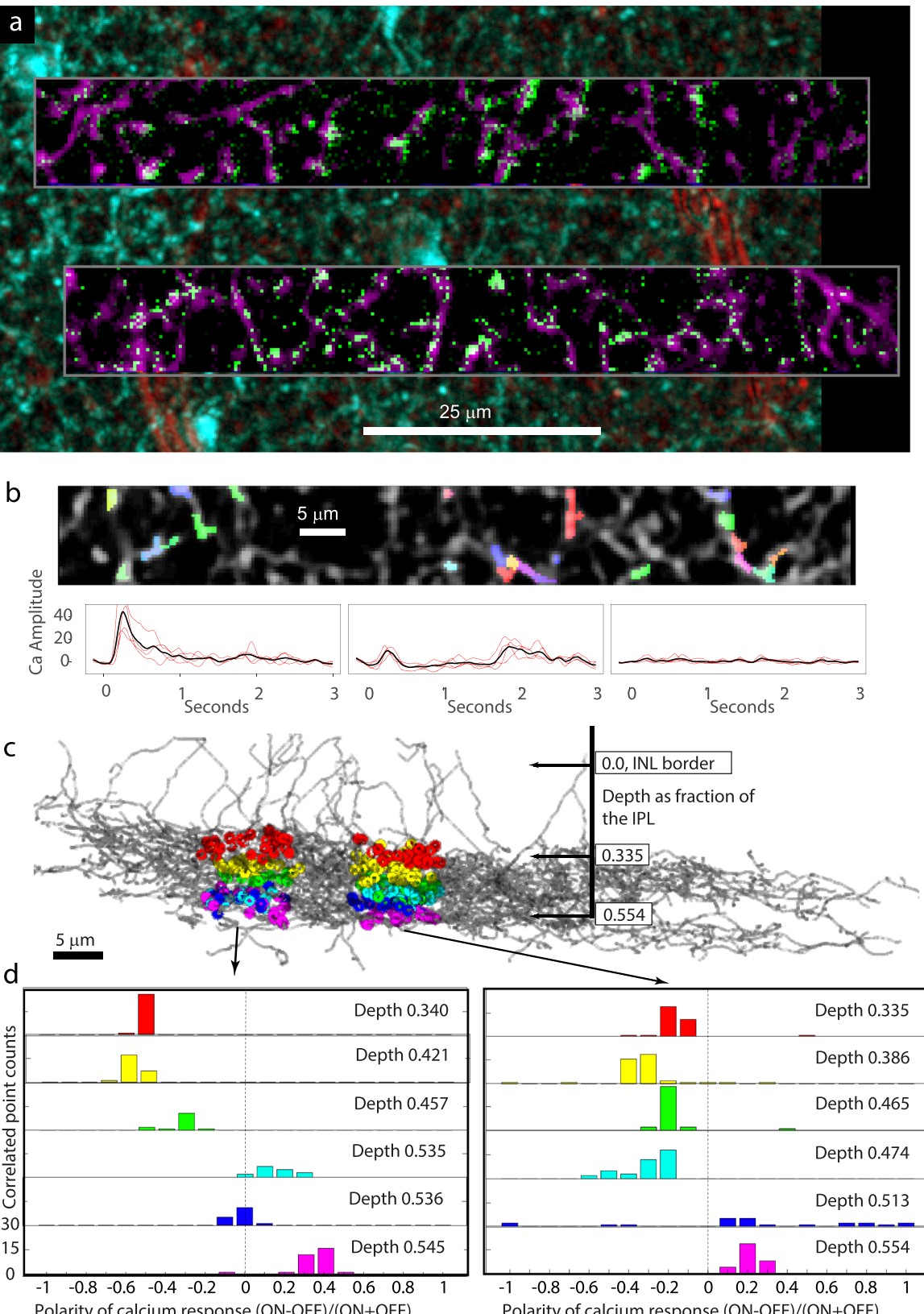

**Fig. 1 | VG3 plexus calcium responses from the retina that was later reconstructed with electron microscopy. a** Two-photon imaging of GCamp6f expression in VG3s. Two-photon structural map of VG3 plexus shown in cyan. Transmitted light image of blood vessels in red. Averages of functional images of VG3 dendrites are shown in magenta. Single frame of calcium response to stimulus is shown in green. **b** Colors indicate pixels pooled for measuring calcium response at targeted points in VG3 plexus (grey). Traces show examples of four raw (red) and averaged (black) fluorescence changes in response to bar stimulus. **c** Colored dots show functionally characterized dendrites that could be mapped onto the 3DEM reconstruction of the VG3 plexus (gray arbors). **d** Histograms show ON/OFF polarities of neurites from each functional imaging plane. Colors correspond to the planes in (**c**). INL inner nuclear layer, IPL inner plexiform layer.

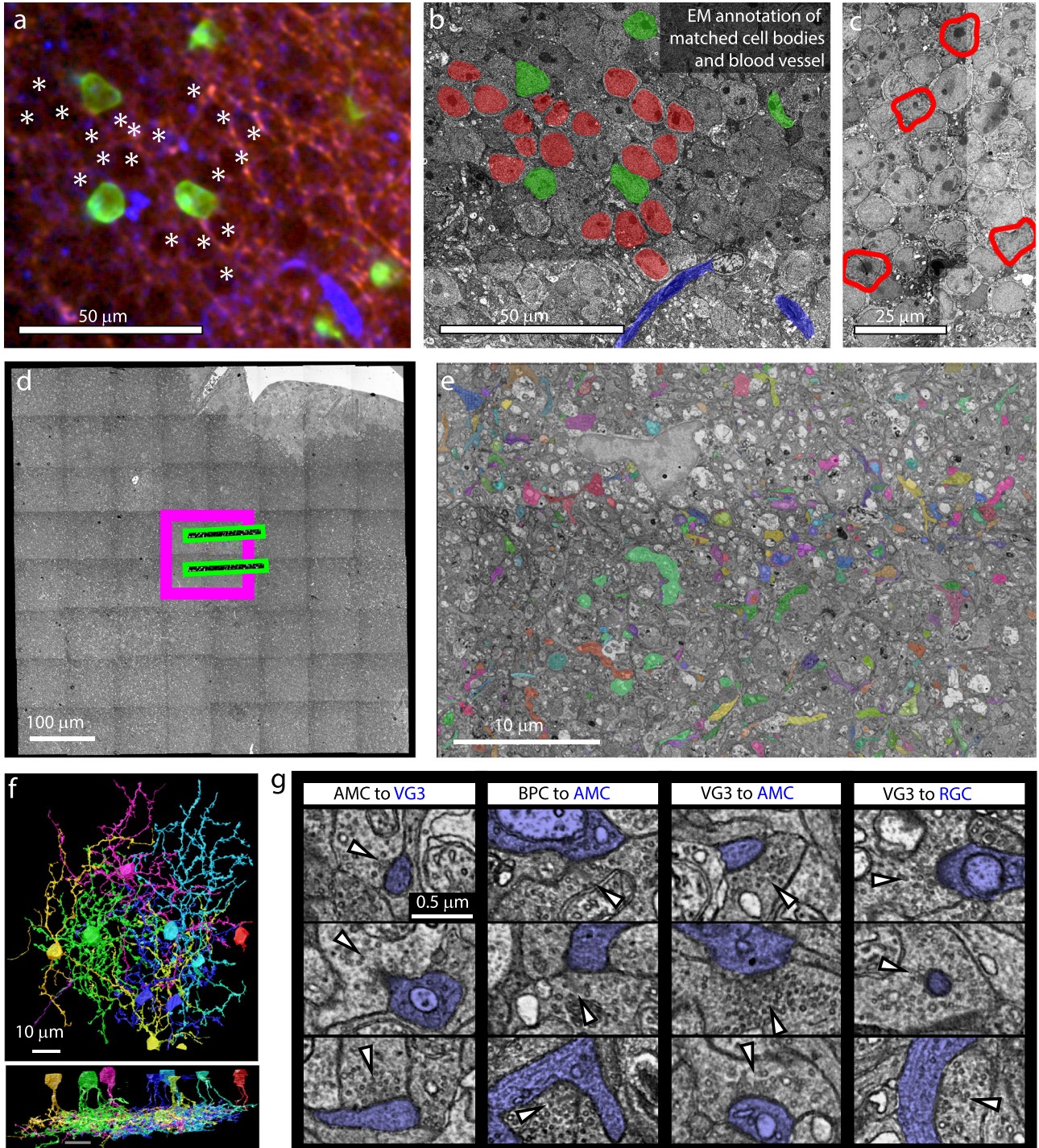

**Fig. 2 | EM imaging and tracing of VG3s in the retina used for correlated light and electron microscopy. a** Fixed 3D confocal image of GCamp6f expression in VG3s (green), aldehyde autofluorescence showing cell nuclei (red), and reflected light imaging of blood vessels (blue). **b** Medium-resolution EM of VG3 nuclei (green), other nuclei (red), and blood vessels (blue) matched to features in (**a**). **c** VG3s (red outline) exhibited a chromatin distribution that was unusual for nuclei in the inner nuclear layer. **d** Relative size and position of 20 nm EM volume (full panel), 4 nm EM volume (magenta box), and functional imaging (green boxes). **e** Example image showing manual tracing of VG3 circuit. Functionally imaged region (top) is marked by expanded vacuoles. **f** Top down and side renderings of reconstructed VG3s. **g** Examples of synapses identified in 4 nm 3DEM volume. Postsynaptic cell is labeled blue. Vesicle cluster is indicated by white arrow. AMC Amacrine cell, BPC Bipolar cell, RGC Retinal ganglion cell.

avoid branching within one another's territories[25]. Often this tiling rule was expressed as branches terminating at the site of membrane contact with a bipolar cell axon of the same type (Fig. 4c).

Ribbon synapses onto reconstructed VG3s were formed by bipolar cells of type 3a (52.3 ± 1.6%), 3b (8.1 ± 1.8%), 4 (9.9 ± 2.4%), 5o (15.8 ± 1.8%), 5i (6.1 ± 2.5%), 5t (3.5 ± 1.4%), xbc (3.1 ± 0.9%), and 6 (1.0 ± 0.6%) (Fig. 4d). Therefore, 65.3% (±2.1%) of the ribbon inputs came from OFF-bipolar cells while the other 34.7 ± 2.1% came from ON-bipolar cells. This sampling of types corresponds to the bipolar cells whose axonal arbors stratify at the same depth as the VG3 plexus.

**Table 1 | Arbor sizes, synapse numbers, and synapse densities for the six most extensively reconstructed VG3s**

| Cell ID# | arbor μm | total syn density syn/um (#) | BC input syn/um | AMC input syn/um | AMC output syn/um | RGC output syn/um | total input syn/um | total output syn/um |
|---|---|---|---|---|---|---|---|---|
| 2 | 1632 | 0.482 (787) | 0.07 | 0.207 | 0.092 | 0.105 | 0.278 | 0.197 |
| 3 | 1526 | 0.37 (564) | 0.074 | 0.155 | 0.064 | 0.065 | 0.229 | 0.129 |
| 4 | 1350 | 0.359 (485) | 0.059 | 0.137 | 0.098 | 0.056 | 0.196 | 0.153 |
| 5 | 807 | 0.304 (245) | 0.036 | 0.13 | 0.077 | 0.051 | 0.166 | 0.128 |
| 13 | 1223 | 0.474 (580) | 0.061 | 0.213 | 0.119 | 0.07 | 0.273 | 0.189 |
| 14 | 918 | 0.37 (340) | 0.07 | 0.17 | 0.076 | 0.05 | 0.24 | 0.126 |
| All | 1243 | 0.393 (500) | 0.062 | 0.169 | 0.088 | 0.066 | 0.230 | 0.154 |
| SE | 134 | 0.029 (78.3) | 0.006 | 0.014 | 0.008 | 0.008 | 0.018 | 0.013 |

Arbor length reflects skeletonized arbor. Bottom row shows mean and standard error for the six VG3s.

Other than type 6, we observed no synapses from bipolar cells stratifying outside the main VG3 plexus (type 1, type 2, type 7, or type 8/9 bipolar cells) (Fig. 4e).

These results argue for a non-zero probability of bipolar cell terminals forming synapses anywhere they overlap with VG3s. The large number of type 3a inputs also suggests a preferred synaptic partnership. We used a Monte Carlo analysis to determine how much this preference deviated from indiscriminate synapse formation. In this analysis, the observed number of synapses between bipolar cells and VG3s were randomly reassigned to bipolar cell types. The probability of synapses being assigned to a given type was determined by the degree of stratification overlap between the bipolar cell type (based on EyeWire[26]) and our VG3 plexus (Fig. 4e).

We found that, in 99.9% of the Monte Carlo results, type 3a bipolar cells were responsible for between 7.5 and 17.1% of bipolar cell synapses onto VG3s (Fig. 4f). In the actual reconstruction, type 3a bipolar cells contribute 53.2% of the synapses between bipolar cells and VG3s, at least three times the proportion expected by chance. The VG3 preference for type 3a bipolar cells appears to reflect both the number of 3a bipolar cells innervating the VG3 plexus (25 3a bipolar cells) and the number of synapses being formed between each type 3a bipolar cell and the VG3 plexus (9.1 ± 1.3) synapses per 3a bipolar cell (Supplementary Table 2). The next most common input type, 5o, formed less than half as many synapses per bipolar cell (18 cells, 3.2 ± 0.5 synapses/cell).

Type 5 bipolar cells include types that differ in the speed of their responses[27,28]. Distinguishing between type 5i, 5o, and 5t ON-bipolar cells anatomically is difficult, and there may be errors in our classification. However, we observe multiple type 5 bipolar cells that are presynaptic to the VG3 plexus in the same territory (Fig. 5b, Supplementary Fig. 4a). The dense overlap argues that VG3s receive some synapses from each variety of type 5 bipolar cells.

## VG3s selectively innervate some transient RGC types while avoiding other RGCs

We determined which RGC types VG3s innervate by reconstructing the arbors of RGCs postsynaptic to VG3. Larger diameter RGC dendrites that left the high-resolution volume were identified in the larger, medium-resolution volume (Fig. 2d) allowing us to find cell bodies and major processes outside of the limited view (~100 μm) of our high-resolution volume. We compared our EM reconstructions of RGCs to the Eyewire Museum[24] as we did with bipolar cells. This process was aided by visualizing the overlap of stratified RGC dendrites with different types of bipolar cell terminals (CellNav).

We were able to identify the types of 47 RGCs innervated by 378 synapses from the VG3 plexus (Fig. 5a). These RGCs included 19 different types, with two-thirds represented by only one or two examples (Supplementary Table 2, Fig. 5a). Three synapses innervated an RGC whose stratification was consistent with an M3 RGC, but whose reconstruction is too fragmentary to identify confidently. Because of

our limited sampling of the large number of total RGC types (42+[29]), it is not possible to rule out any potential VG3-RGC combination using this dataset.

Much of the VG3 to RGC innervation targeted RGCs with small to medium ON/OFF receptive fields that are sensitive to moving stimuli. These included 5ti RGCs (W3, UHD, small-field transient ON/OFF, 19.8%, ±2.27%), type 63 RGCs (F-mini-ON, 11.6%, ±1.61%), and type 37 RGCs (ON/OFF direction-selective, 12.6%, ±1.74%). Almost a third of these synapses went to three monostratified transient OFF RGCs that stratify at the same depth within the IPL: 4i RGCs (7.7%, ±1.04%), 4on RGCs (6%, ±1.35%), and 4ow RGCs (transient OFF Alpha RGC, 15.7%, ±1.77%). The prevalence of VG3 synapses to the 5ti and 4ow RGCs is consistent with physiological, genetic knockout, and behavioral evidence[30] that the VG3 innervation of these two RGC types mediates the detection of looming stimuli, a major function of VG3 cells. VG3s also heavily innervated the type 6sw monostratified transient ON RGCs (medium field transient ON, 13.9%, ±2.65%).

We found very few synapses innervating RGCs with sustained responses. The type 8w (sustained ON Alpha, 0.2%, ±0.24%) was the only sustained RGC recipient we observed. The few synapses onto the 8w were formed by one dendrite that strayed outside of the main VG3 plexus and which was also unusual in being predominantly innervated by type 6 bipolar cells (Supplementary Fig. 5a).

We were surprised to find little innervation of type 28, 72, or type 73, Suppressed-by-Contrast RGCs. Previous studies have shown synaptic connections between VG3s and type 73 RGCs[30] and have demonstrated that VG3s inhibit type 73 RGCs and are important for their characteristic reduced firing in response to local visual stimuli[31]. We found only one likely type 73 receiving only 5 synapses (1%, ±0.44% of total output to RGCs). The stratification of type 73s (primarily outside of the Chat bands[32]), may provide few opportunities to form synapses relative to other target RGC types. We note however, that one of these synapses was exceptionally large (Supplementary Fig. 5). Maximizing the output at individual synapses, therefore, may compensate for limited opportunities to form synapses with Suppressed-by-Contrast RGCs.

To determine the extent to which stratification overlap could explain the RGC innervation we observed, we repeated the stratification-based Monte Carlo simulation for RGC types (Fig. 5d). A potential source of bias in comparing a depth-dependent Monte Carlo model of synapse formation to the observed data is that the formation of a real synapse between the VG3 plexus and a particular RGC arbor means that there is an increased probability of nearby opportunities for more synapses to be formed between the same pairing. To reduce this bias, we pruned observed synapses so that synapses between the same pair of cells were at least 10 μm away from each other. The type 37 (ON/OFF direction-selective cells) stood out as forming many more synapses than predicted by overlap (observed = 40 synapses, predicted 95 CI = 1−9). These cells stratify at the cholinergic bands in the IPL that define the outer boundaries of VG3 stratification yet are a primary target for VG3 innervation.

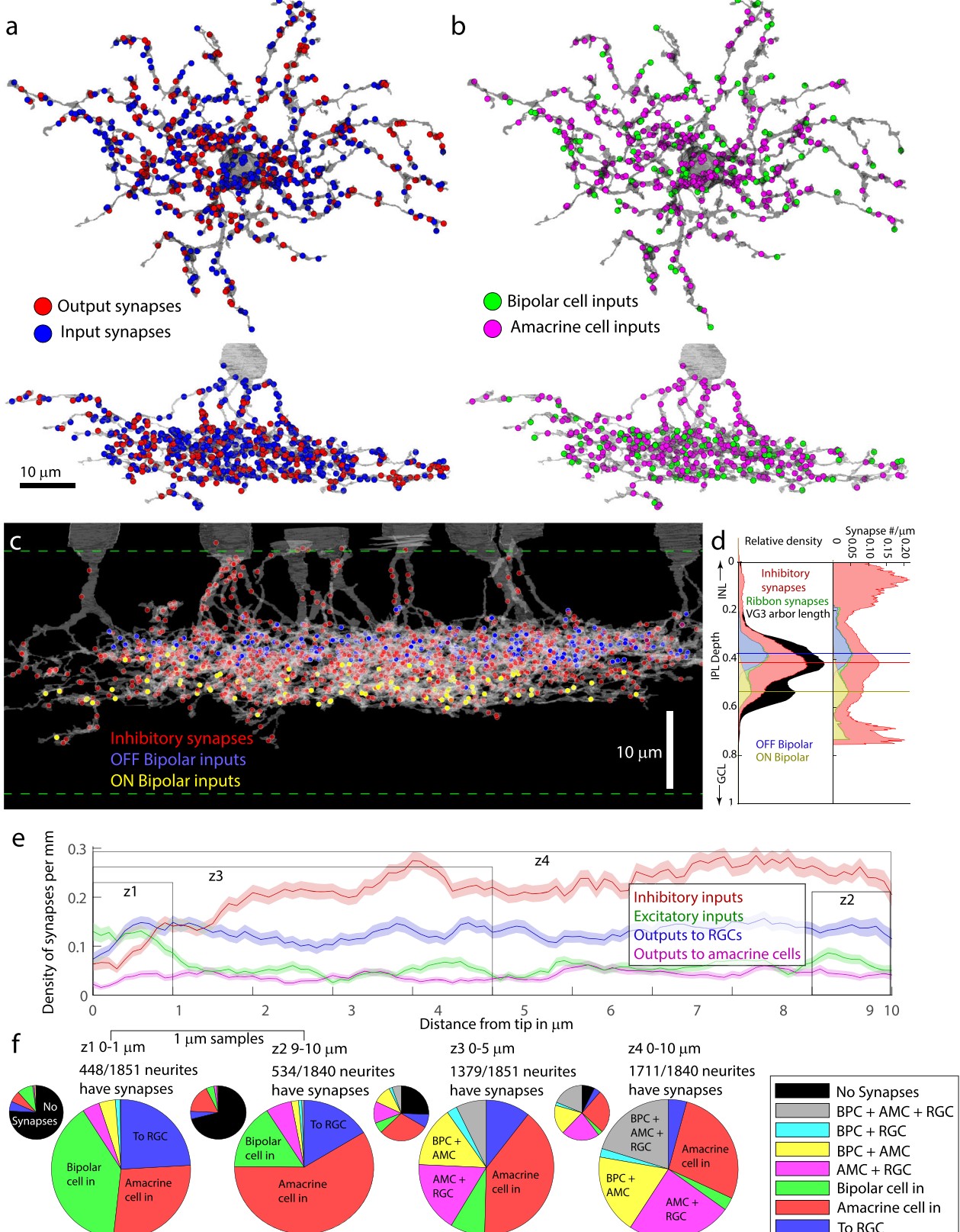

Fig. 3 | **Distribution of synapses types across VG3 arbors of the reconstructed volume. a** Mixing of input (red) and output (blue) synapses on VG3 arbor. Top down view is shown above the side view. **b** Distribution of bipolar cell (green) and amacrine cell (magenta) inputs on VG3 arbor. **c** Distribution of synapses from amacrine cells (red), OFF bipolar cells (blue), and ON bipolar cells (yellow) across the depth of the VG3 plexus. **d** Histograms showing the distribution of synaptic input across the depth of the VG3 plexus. Left = absolute synapse number.

Right = synapse density relative to VG3 arbor length. **e** Mean density of synapses relative to distance to tip of VG3 dendrite. Shaded region shows standard error for distance bin of 1840 dendrite tips. Z# indicates zones shown in (**f**). **f** Frequency of different synapse combinations found within dendrite zones shown in (**e**). Small pie chart includes dendrites with no synapses in black. AMC Amacrine Cell, BPC Bipolar Cell, INL Inner Nuclear Layer, IPL Inner Plexiform Layer, GCL Ganglion Cell Layer, RGC Retinal Ganglion Cell.

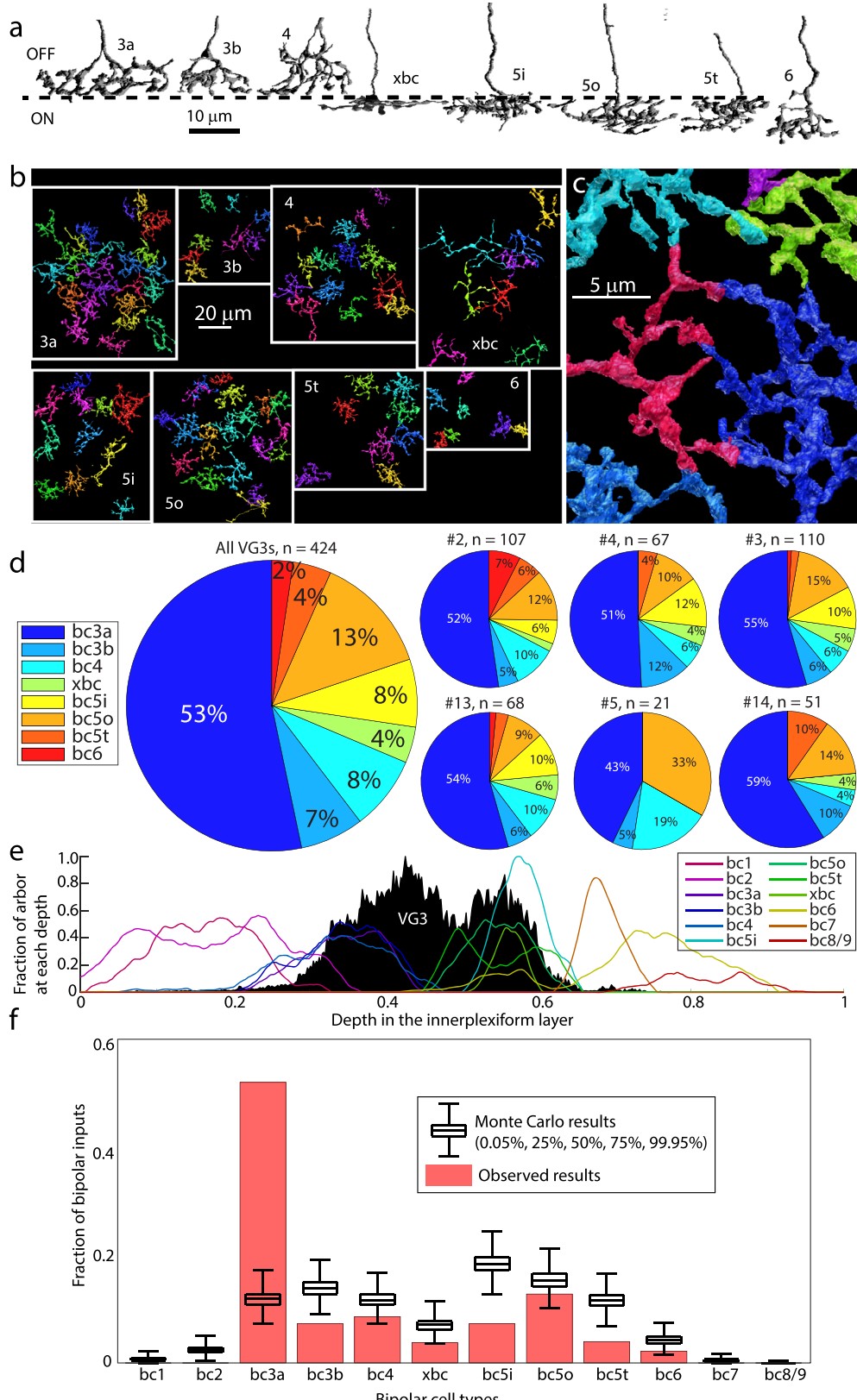

**Fig. 4 | Bipolar cell types innervating VG3s. a** Side view of example of each bipolar cell type innervating VG3 plexus. **b** Top down views of bipolar cell types innervating VG3 plexus. **c** Example of terminal homotypic touches that helped to group bipolar cells of the same type together. **d** Percent of bipolar input from each bipolar cell type innervating VG3 plexus (large pie chart) and each individual VGC (small pie charts). **e** Depth distribution of bipolar cell types according to Eyewire (colored traces) relative to our reconstructed VG3s (black histogram). **f** Observed number of each type of bipolar cell synapses innervating VG3s (red) relative to Monte Carlo prediction (black) that was based on the arbor overlap in (**e**) and data in (**d**) (424 synapses from 6 VG3s).

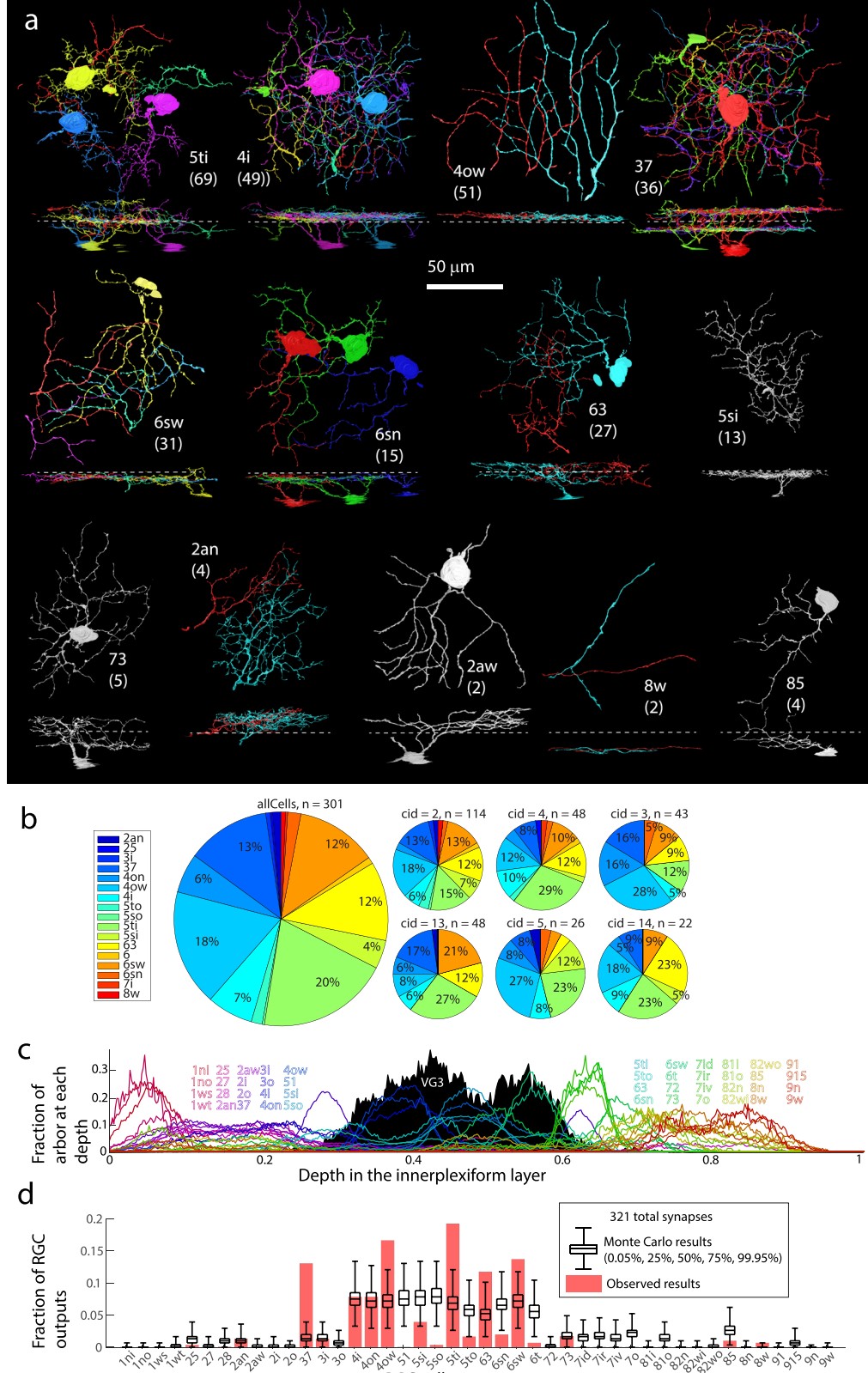

**Fig. 5 | RGC types innervated by VG3s. a** Top down and side views of most RGC types innervated by VG3 plexus. Numbers in parentheses denote number of VG3 synapses to RGC type, and dashed line indicates the approximate boundary between the ON and OFF sublamina of the IPL. **b** Percent of VGC-RGC synapses targeting each RGC type overall (large pie chart) and for each individual VGC (small pie charts). **c** Depth distribution of RGC types according to Eyewire (colored traces) relative to our reconstructed VG3s (black histogram). **d** Observed number of VG3 synapses (red) innervating each RGC type relative to Monte Carlo prediction (black) based on the arbor overlap in (**c**). RGC Retinal Ganglion Cell.

To estimate the fraction of excitatory input VG3s contributed to their preferred partners, we counted bipolar inputs across the available arbors of four RGCs types (see also Fig. 7b–e). We found VG3s constituted 27.9% (36 VG3 inputs vs 93 bipolar cell inputs) of excitatory inputs onto 4ow, 38.4% (38 vs 61) onto 5ti, 22.7% (22 vs 75) onto the 37, and 20.6% (21 vs 81) onto the 6sw. Thus, while VG3 inputs constituted a minority of the inputs (mean of the 4 types = 27.4 ± 4.0%), their frequency was on the same scale as bipolar inputs.

### Comparisons of anatomical models and functional recordings indicate a short length constant for VG3 signal spread

We next tried to determine the distance over which individual bipolar cell inputs influenced the VG3 plexus. We used calcium imaging of the same tissue and other VG3s to constrain a simple model for exponential decay ($e^{-d/\lambda}$). We first calculated node-edge skeletons of the VG3s (internode length of -0.1 μm) so that we could measure the linear distance (d) between any bipolar cell input and every other part of a VG3 arbor.

We first estimated the VG3 length constant by comparing predictions of the EM reconstruction to previously published measurements of the ON/OFF polarity of the VG3 plexus[8]. Polarities (−0.18, −0.44, −0.45, −0.32, 0.02, 0.22, 0.23) were reported for seven IPL depths spanning the VG3 plexus. To make our anatomical prediction of polarity, we assigned each ribbon synapse a polarity based on the reconstruction of the bipolar cell (ON or OFF). We then tested a range of length constants (1:150 μm) for exponential decay. For VG3 neurites whose depth matched the published values (Supplementary Fig. 6a), we compared the predicted ON/OFF polarity to the published calcium polarity. The best-fit length constant of 19 μm (Fig. 6a) predicted a shift of polarity with depth (−0.31, −0.31, −0.35, −0.30, −0.22, 0.19, 0.23) similar to previous calcium recordings.

We next estimated the length constant by comparing our anatomical predictions of ON/OFF polarity to the polarities derived from functional imaging of the same dendrites (Fig. 1c, d, Supplementary Fig. 6b). Before comparing anatomical and functional predictions of polarity, we used our EM reconstructions to characterize the reliability and filter our functional recordings. The EM reconstructions allowed us to identify points within the functional recordings that should be electrotonically close enough to one another that their calcium ON/OFF polarity should be very close together. Comparing these points identified one plane that was responsible for roughly half the close neighbor prediction error (average polarity index difference to neighbor = 0.214 with and 0.122 without). This plane was removed from subsequent analysis. The remaining light/EM correlation points were grouped together if less than 1 μm apart reducing the initial 334 points to 124.

We estimated a best-fit length constant using an exponential decay model that included simulating a range of added noise and ON/OFF input scaling factors. The length constant that produced the best overall fit for the functional ROIs was 10.2 μm with a mean polarity error of 0.202 (Fig. 6a). A slightly higher error level of 0.22 includes length constants ranging from 4.8 to 22 μm. To better understand the reliability of this prediction, we tried fitting our model to randomly scrambled functional polarities and found a best length constant of 71 μm with an average error of 0.258. While the real fitting performed better than random, it did not approach the best possible fit (close neighbor error = 0.12). The best-fit length constant also differed from the length constant estimated from the comparison to the previously reported polarities (19 μm). A possible source for the range of length constant results in the polarity fitting could be due to noise in our functional measurement of polarity or incorrect assumptions in how ON and OFF bipolar signals mix in VG3s. We, therefore, used a third-length constant estimation that did not use response polarity.

We calculated correlation coefficients for the calcium signal (ignoring stimulus timing) of pairs of correspondence points (Supplementary Fig. 6c). For this calculation, we only compared correspondence points in the same imaging plane that were likely to have similar ON/OFF polarities. We next calculated an EM-based bipolar cell influence correlation coefficient for each pair of correspondence points. This correlation was calculated using the vector of the influence of each bipolar cell for each correspondence point. We found that an exponential decay length constant of 18 μm produced the best match in correlation coefficients between functional recordings and anatomical predictions (Fig. 6a).

Considering the three estimates, we settled on a functional length constant of 16 μm (average of 10.2, 18, and 19). Figure 6b shows the distribution of estimated VG3 polarity across IPL depth using this length constant and predicts a shift from OFF-biased responses (OFF sublamina) to a more even balance of ON and OFF responses (ON sublamina) that matches the calcium imaging for these cells (Fig. 1d) and previous physiological characterizations of VG3s[8]. Interestingly, the strong OFF bias of the VG3 arbor in the OFF sublamina was not due to a balanced switch between ON and OFF influence but a doubling of OFF influence compared to a relatively unchanged ON influence (Supplementary Fig. 7).

This length constant model does not consider differences in dendrite diameter, channel distribution, timing, or other factors important to dendritic computation. Notably, this model does not account for the influence of inhibitory amacrine cell synapses dividing the ON/OFF strata. These amacrine inputs not only change membrane conductance but can shape responses in a feature-selective manner (i.e., alter polarity or correlation coefficients of dendrites). However, the simple length constant model nonetheless provides a realistic baseline for estimating the functional relationship between bipolar cell inputs and the response of VG3 neurites. In particular, the calcium rises constraining the model are tied to the neurotransmitter release that the model attempts to predict.

### Electrotonic modeling suggests inhibitory shunting can preserve biases in neurite polarity

To investigate possible mechanisms for the short calcium polarity length constant, we ran NEURON[33] simulations of the VG3 plexus. The model included the skeletonized VG3 arbors, the diameter of each segment, and the location of bipolar cell and amacrine cell inputs. In these simulations, we varied four cell properties: passive membrane-specific conductance (gm), excitatory synapse conductance (Ge), inhibitory synapse conductance (Gi), and a scaling factor to reduce inhibition during ON vs. OFF stimulation (OnScale). We varied the OnScale because we do not know which inhibitory synapses are active during ON or OFF activation, but previous work suggests that inhibition during ON stimuli is -50% of that recorded during OFF stimuli[5]. For each simulation, we activated all OFF bipolar cell inputs and all inhibitory synapses. We then activated all ON bipolar and inhibitory synapses (inhibition multiplied by the OnScale factor). We assigned the ON/OFF polarity of neurites as the maximum voltage change during ON stimulation minus the maximum change in voltage during OFF stimulation divided by the total change in the voltage ((ON − OFF)/(ON + OFF)).

We started by testing a wide range of conductances, including values outside of the realistic range for VG3s. We found the best-fit conditions without inhibition (gm = 10^−2 nS/cm^2, Gi = 10^2 nS, polarity error = 0.22, corrcoef = 0.628) performed almost as well as the best fit with inhibition (gm = 10^−3 nS/cm^2, Ge = 10^3 nS, Gi = 10^0 nS, OnScale = 1, polarity error = 0.21, corrcoef = 0.603). However, both these sets of conditions include passive membrane conductances and excitatory conductances that are outside realistic ranges for VG3s. In particular, the high passive conductance would result in a median electrotonic length constant of 35 μm for our cells (median diameter = 0.5 μm). This short electrotonic length constant allows the model to produce the range of ON and OFF polarities we observe in VG3s.

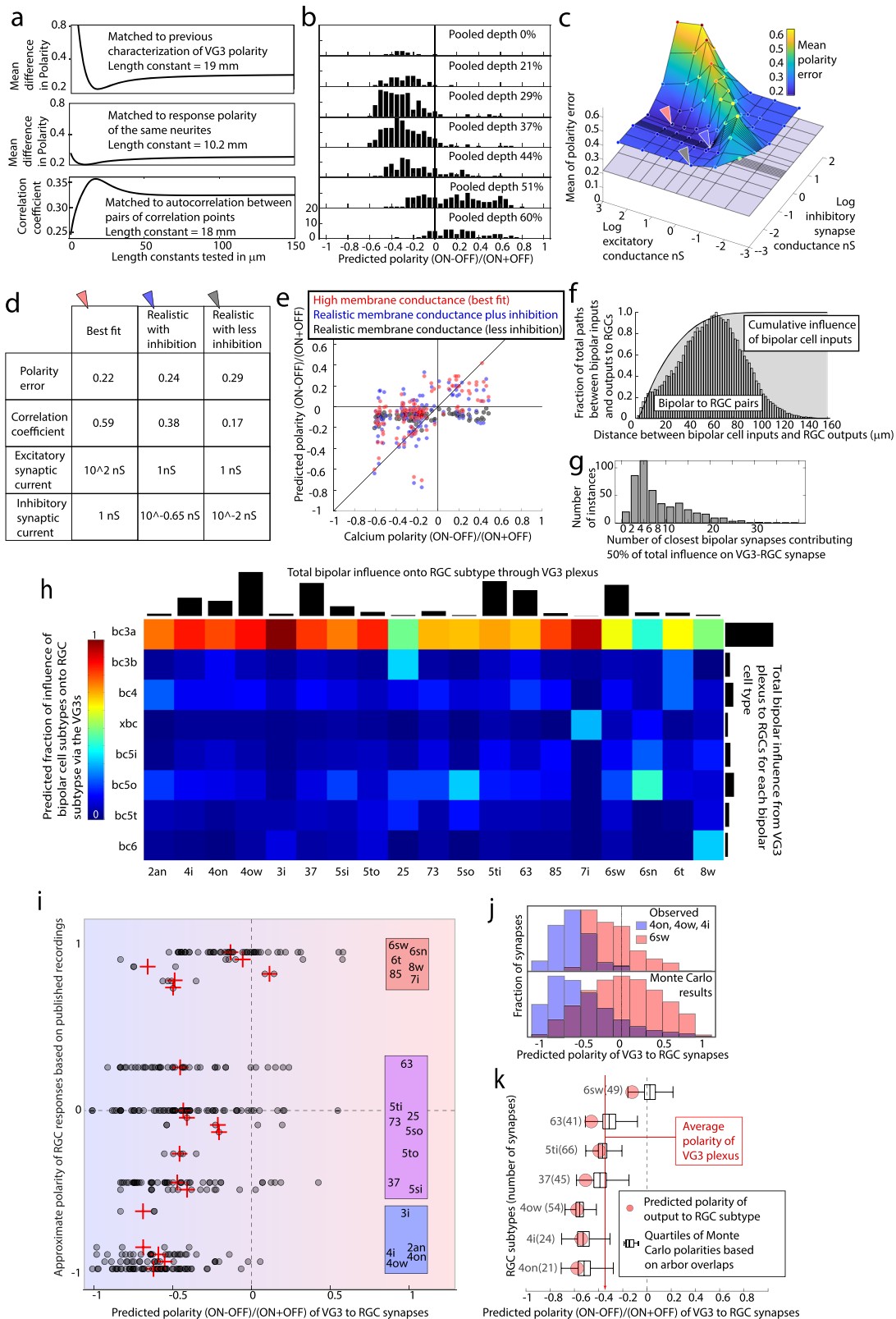

We next identified the best fit conditions given membrane and synapse conductances more consistent with previous recordings of VG3s[3,5] (gm ≤ 10^−4 nS/cm^2, Ge ≤ 10^0 nS). These conditions correspond to a median electrotonic length constant of at least 354 μm. The best realistic fit was gm = 10^−4 nS/cm^2, Ge = 10^0 nS, Gi = −10^0.65 nS, and OnScale = 0.5 (Fig. 6c−e, polarity error = 0.25, corrcoef = 0.38). If we remove inhibition under these conditions, the match between the model and physiological recordings decreases dramatically (Fig. 6c−e, polarity difference = 0.2885, corrcoef = 0.170). Without inhibition, the 354 μm electrotonic length constant produces a uniform ratio of ON and OFF responses across each VG3 arbor (Fig. 6e). The reduction of excitatory signal spread via inhibitory shunting, therefore, appears to be a plausible mechanism for producing the diversity of ON and OFF responses observed in VG3s.

**Fig. 6 | Modeling bipolar to RGC transmission through the VG3 plexus.**
**a** Predicted polarities using a range of length constants compared to recorded
response polarities estimated a functional length constant of 10–20 μm.
**b** Distribution of predicted VG3 response polarity using a length constant of 16 μm.
Each histogram represents a different depth in the IPL. **c** Surface plot of polarity
prediction errors of NEURON models of VG3s in which specific membrane con-
ductance is 10^−4 nS and OnScale is 0.5. Arrows indicate three combinations of
excitatory and inhibitory synaptic conductances that are shown in (**d**, **e**). **d** Table
showing conditions and model accuracy for the three points with matching arrows
indicated in (**c**). **e** Plots of calcium response polarity vs. NEURON predicted ON/OFF
polarity. Colors indicate the three model conditions indicated in (**c**). **f** Distribution
of distances between bipolar inputs and VG3 outputs to RGCs. Histogram shows
distribution of raw distances. Gray curve shows the fraction of total predicted
bipolar cell influence on RGC outputs included within each distance. **g** Distribution
minimum number (closest) of bipolar cell inputs predicted to provide 50% of the

drive to each VG3-to-RGC synapse. **h** Adjacency matrix showing predicted influence
of each bipolar cell type on each RGC type via the VG3 plexus. Each column (RGC
type) is normalized to sum to one. Black bars indicate the total influence estimated
for each RGC (top) and bipolar cell type (right). **i** Distribution of predicted polarity
of VG3-to-RGC synapses (gray circles, means in red crosses) relative to the pub-
lished polarity of each RGC type. Blue indicates OFF bias. Red indicates ON bias.
Purple indicates mixed ON and OFF responses. **j** Top: Difference in predicted
polarity of synapses innervating monostratified OFF and ON RGCs (99 ON synapses
and 49 OFF). Bottom: Stratification-based Monte Carlo redistribution of synapse
positions produces similar polarities as top panel (99 ON synapses and 49 OFF).
Overlap of distributions is indicated by purple. **k** Stratification-based Monte Carlo
redistribution (black plots) of synapses from (**i**) broken down by major targets of
VG3 innervation. Predicted polarities from actual synapse locations are shown in
red. Whisker plots indicate quartile boundaries.

In A17 amacrine cells, functional independence between different
parts of the arbor is produced, in part, by the narrow diameters of
linking neurites[34]. We found that smoothing the VG3 arbors to match
the overall median diameter did not dramatically change the match
between observations and predictions at the above best-fit conditions
(gm 10^−4 nS/cm^2, Ge 10^0 nS, Gi 10^0.65 nS, OnScale 0.5, mean error
0.25, cc 0.42). Changes in neurite diameter, therefore, do not appear to
be critical for producing the observed neurite polarization.

**Functional connectomic predictions of VG3 influence on RGCs**
We used the influence length constant estimated above (16 μm) to
predict which bipolar cell types drive calcium responses in the VG3
neurites connected to each RGC type. We found that, for the typical
VG3-to-RGC synapses, about half of their bipolar cell drive originated
from half a dozen or so bipolar synapses that occurred within 21 μm of
the output synapse (Fig. 6f, g). Summing up all the bipolar cell influ-
ences, we found that type 3a bipolar cells dominated the estimated
responses presynaptic to most RGC types (Fig. 6h). Deviations were
observed only in types for which the sampling was low.

We tried to determine if the predicted response polarity of VG3
neurites could also predict which type of RGC the neurite would
innervate. We compared the predicted polarity of the VG3 neurites
presynaptic to RGCs to the published response polarity of the RGC
types (Fig. 6i[29]). The response polarities of most RGCs listed are
expected to be primarily driven by the bipolar cell types that innervate
them directly. The correlation coefficient between the predicted
polarity of VG3 neurites at each synaptic connection to RGCs (synapses
to Suppressed-by-Contrast RGCs excluded) and the polarities of their
RGC target types was only 0.33 (95 CI 0.22–0.42). Notably, the varia-
tion in the polarity of neurites innervating each cell type is large rela-
tive to the difference between RGC types. Monostratified ON cells
(6sw, 6sn) receive some VG3 synapses from neurites biased towards
ON responses (-0.5 polarity), but also many synapses from neurites
biased towards off responses (-−0.5, Fig. 6i). However, the model
predicts enough distance between the synapses innervating mono-
stratified OFF RGCs and monostratified ON RGCs that the overall
polarity of the VG3 neurites innervating them is distinct (Fig. 6k, 99
OFF synapses, SE range from −0.61 to −0.57 vs. 49 ON synapses SE
range from −0.16 to −0.08, 95% difference range 0.39–0.60, see
Methods for bootstrap calculation of difference range).

To determine if the depth of arbor stratification was sufficient to
explain the difference in VG3 neurite polarity presynaptic to ON and
OFF RGCs, we performed a Monte Carlo analysis where VG3-to-RGC
synapses were moved to different locations on the VG3 arbor. In
choosing the new location of the synapse, each node on the VG3 arbor
was weighted by its overlap with the depth profile of the associated
RGC type (Eyewire, Fig. 5c). We found that the VG3 neurite polarities of
the randomized synapses were similar to those of the observed
synapses (10,000 iterations, Fig. 6k). The real 95% difference range for

neurites innervating monostratified ON and OFF cells (0.39–0.54)
closely matched the 95% range of Monte Carlo values (0.35–0.64).
Therefore, the difference in RGC type stratification appears to be
sufficient to produce most or all the differences we predict in the
polarity of VG3 neurites innervating different types of RGCs.

**Indirect and direct bipolar input input to RGCs**
We next asked whether RGCs receive direct innervation from the same
bipolar cells that drive the response of the VG3 neurites that innervate
them (BC-VG3-RGC triad motif). While reconstructing RGCs, we
annotated synapses from bipolar cells that also innervated the VG3
plexus. We also used an apposition detection algorithm to perform a
more systematic search for such synapses on four individual RGCs
representing some of the most frequently innervated RGC types (4ow,
5ti, 37, 6sw). Altogether, we identified 460 connections between
bipolar cells that innervate the VG3 plexus and RGCs innervated by the
VG3 plexus. These connections were produced by 440 ribbon synap-
ses (some ribbons innervated multiple RGCs).

Most RGC types innervated by VG3s also shared some bipolar cell
input with them (Fig. 7a). In the four patches of RGCs in which all
bipolar cell inputs were recorded, we found that shared inputs con-
stituted 48.4% (4ow, 45 of 93), 68.9% (5ti, 42 of 61), 60.0% (37, 45 of 75),
and 54.3% (6sw 44 of 81) of the bipolar cell input in these regions
(Fig. 7b–e). Given that not all VG3s and associated bipolar cells in the
region have been reconstructed, these numbers represent a lower
bound. The mix of bipolar cell types that the reconstructed RGCs
connected to directly (strictly matched to stratification) was different
from the mix we predict would influence them through the VG3s
(heavily biased towards 3a, Fig. 7a vs. Fig. 6h).

Given the ubiquity of the BC-VGC-RGC triad connectivity, we
asked if this circuit motif reflects the kind of synapse-specific triad
motif observed in the dorsal lateral geniculate nucleus (dLGN)[35–37] and
other retinal circuits[38,39]. That is, are the synapses that constitute the
triad clustered together within a micrometer or so of each other? We
found that the connections between the bipolar cells and the VG3 and
RGC were often from the same ribbon synapse. Consistent with other
ribbon synapse dyads in the retina, each ribbon synapse innervating a
VG3 also innervated one (222 of 259) or two (37 of 259) other neurons.
The non-VG3 cell innervated by 48% of these ribbon synapses was an
RGC. Of these RGC innervating ribbons, 89.5% (119 of 133) innervated
RGCs that were also known to be innervated by the VG3 plexus (Fig. 7f).
Thus, the observation that the same ribbon synapse innervated a VG3
and an RGC was a strong predictor that the VG3 would also innervate
the RGC. However, the synapses connecting the VG3 to the RGC did
not seem especially targeted to the location of the ribbon dyad. The
median distance between the bipolar-VG3 input and VG3-RGC output
for all triad motifs was 13.0 μm with only 5.6% under 1 μm (Fig. 7g).
Therefore, most BC-VG3-RGC triad motifs appear to operate at a local,
but not synapse-specific, level of feedforward integration.

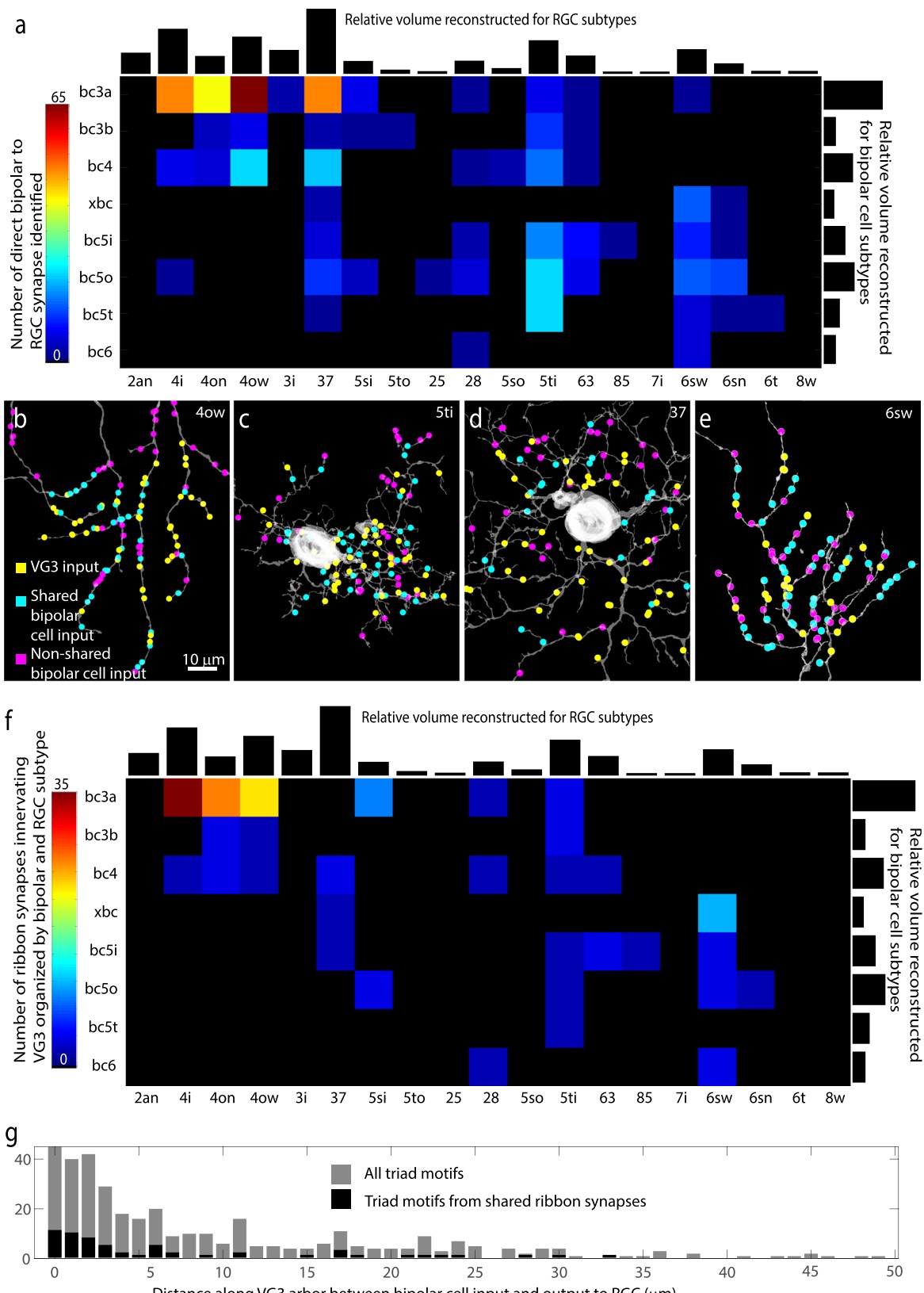

**Fig. 7 | Quantification of BC-VG3-RGC triad motifs. a** Matrix showing direct synaptic connections between bipolar cell types and RGC types. All cells included also synapse with VG3s. Black bars show relative volume of segmented voxels for RGCs (top) and bipolar cells (right). **b−e** Shared, non-shared, and VG3 inputs onto four RGCs. **f** Matrix showing counts of ribbon synapses innervating synaptically connected VG3 and RGC. Black bars show relative volume of segmented voxels for RGCs (top) and bipolar cells (right). **g** Histogram showing counts of triad motifs according to the distance between the BC-VG3 synapse and the VG3-RGC synapse. All motifs pooled are shown in gray. Shared ribbon motifs are shown in black.

## Discussion

### Mapping and representing subcellular pathways

By reconstructing a network of mouse VG3s and their synaptic partners and combining this information with characterizations of the response properties of the VG3 dendrites, we attempted to provide a detailed view of visual information flow through a plexus of mixed input/output dendrites. We found that the response properties of VG3s reflect a spatially biased mixing of their bipolar cell inputs. The short length constant for calcium spread means that different populations of VG3 inputs influence different VG3 output synapses. At the same time, the large fraction of inputs from type 3a bipolar cells means that all output synapses should have a significant component of fast transient OFF drive.

We represented the flow of information through the VG3 plexus using the adjacency matrix in Fig. 6a. We believe that using this adjacency matrix to represent the influence of different types of inputs on different types of outputs is a uniquely useful way to represent the connectivity of a population of cells. In the case of neurons with mixed inputs and outputs, data representations that allow for distinct relationships between different pairs of partners are necessary. However, even for neurons with segregated inputs and outputs, descriptions of connectivity could benefit from data representations that allow for weighted or conditional influences between partners.

### A complementary plexus

While "amacrine" means "no axon", the function of many amacrine cells, such as AII[40], starburst[41], and poly-axonal[42,43] cells depend on input and output synapses being restricted to different parts of the arbor. In contrast, the input and output synapses of the VG3s are evenly intermixed throughout their dendrites. The result is that VG3s generate a plexus that is complementary to the bipolar cells at a given depth and retinal position. The signal is complementary in that: (1) It is partially matched to nearby bipolar cells in terms of spatial position and polarity. (2) It is transmitted to the same RGCs that are innervated by the nearby bipolar cells. (3) The VG3 signal adds selectivity for small moving objects (surround motion inhibition) and looming stimuli (in the OFF sublamina).

Many amacrine cells in the retina[38,39] and local inhibitory neurons in the dLGN[37,44] generate local feedforward triad motifs. These local triad motifs differ from those we describe in VG3s in that they are inhibitory and occur within a micrometer or so of one another[37]. The delayed inhibition of these triads is, therefore, in a position to tune postsynaptic responses to the onset of transmitter release[45,46]. The longer distances between the inputs and outputs of the VG3 triads mean that the output synapse is less likely to be dominated by a single input but will rather integrate a mix of inputs biased to a patch of the IPL. The value added by the VG3 triad is likely this integration of fast, surround-inhibited ON/OFF drive that is still specific to a narrow patch of visual space[8].

### Translating VG3 neurite responses into RGC activity

Our modeling makes predictions about the polarity of calcium responses for VG3 neurites innervating different kinds of RGCs. However, there are several important uncertainties in determining the effect of these neurites on the RGCs they innervate. First, we don't know whether a given VG3-RGC synapse is excitatory or inhibitory. Previous electrophysiology demonstrates that VG3 provides a strong excitatory drive to OFF Alpha and 5ti RGCs[5,6,47] and strong inhibition to Suppressed-by-Contrast RGCs[9]. However, their influence on other RGC types is unclear and whether innervation to a given type is entirely excitatory or entirely inhibitory is unknown. A second uncertainty is the relationship between calcium levels in a VG3 neurite and levels of neurotransmitter release. There is evidence that glutamate release from VG3s is more polarized than the calcium signal in the dendrites[7]. Finally, VG3 inputs constitute only a minority of inputs in even the

most strongly innervated RGC types so that bipolar cells and other amacrine cells may dominate the response profiles of most RGC types connected to VG3s.

Type 6sw RGCs are a type with particularly strong input from VG3s. These cells are reported to have a negligible OFF response[29,48] but were innervated by VG3 neurites that we predict to have a mixed ON/OFF response. The ON/OFF polarity of the VG3 neurites therefore does not seem to directly translate to an excitatory ON/OFF response in these cells. One possibility is that the VG3 input to 6sw RGCs is inhibitory. A second possibility is that the OFF response of the neurites bearing these synapses is not driving synaptic release enough to noticeably compete with other inputs to 6sws. Arguing against the possibility that OFF calcium responses are too weak to drive synaptic release in distal VG3 neurites are recordings from polyaxonal amacrine cells (PAS4/5). These cells are innervated by the most distal (ON sublamina) tips of VG3s and receive a mixed ON/OFF drive from these synapses[49].

The available evidence, therefore, points towards VG3s providing ON/OFF object motion inhibition to the otherwise ON sustained 6sws. However, more direct characterization of the properties of VG3 output synapses and their postsynaptic integration will be required to resolve the influence of the VG3s on the large number of RGC types they innervate.

### Preferences within promiscuity

Overall, we found synapses between VG3s and most of the bipolar cells and RGCs they overlapped with. However, most synapses targeted a small subset of cell types, and the other potential partners constituted a long tail of weak connections. This general pattern is consistent with the results of other connectomic studies[50], in which synaptic preference is expressed as a bias in the probability of synapse formation rather than a binary rule. For a given circuit, it can be difficult to determine if the lack of a clear cutoff in synaptic preference reflects noise in the biology, noise in the observation, or computationally important adaptations. In the case of the bipolar cell input to VG3s, there is a clear ethological role for the non-dominant inputs. Pooled together, the non-primary inputs constitute a significant fraction of the VG3 excitatory drive, which results in a mixed polarity receptive field that shifts with the depth of the IPL. This mixing of transient ON and OFF responses has a clear behavioral consequence as it is required for generating strong VG3 responses to looming stimuli[30].

The functional significance of the strength of connectivity between VG3s and different RGC types appears more varied. The synapses with 8 ws (sustained ONα) were few and appeared to reflect permissive synapse formation in dendrites that wandered out of their usual stratification depth. These synapses appear to be a clear example of weak connectivity as part of biological noise. On the other hand, we also saw only a few synapses onto type 73 (suppressed-by-contrast) RGCs despite the well-established role VG3s play in providing the inhibitory suppression that defines the suppressed by contrast light responses of this cell type[9,10]. We do not know if these synapses are particularly strong (though few), if our sample is unusual in its lack of strong VG3 to 73 synapses, or if there is some additional inhibitory mechanism we are not aware of such as non-synaptic release of glycine.

### Limited signal spread

The VG3 calcium length constant (16 μm) is within the range of what we might expect from an amacrine cell maintaining a degree of local compartmentalization in its arbor. This length constant is consistent with those previously published in retinal amacrine cells (13 μm[34]) and pyramidal cells (7–19 μm[51]). This length constant is far smaller than the plausible length constant of passive electrotonic spread of depolarization. Several factors could limit the spread of calcium signals. First, the relationship between depolarization and intracellular calcium levels is non-linear and can be affected by channel distributions and

calcium release from intracellular stores. Changes in neurite diameter such as those present in VG3s[7] can also limit the spread of calcium. However, our results argue the biggest contributor is likely inhibitory shunting.

Inhibitory shunting is a common mechanism in which the conductivity of inhibitory synapses limits the spread of depolarization driven by excitatory synapses[52]. In our modeling, pairing bipolar cell input excitation with amacrine input inhibition reduced the spread of excitation enough to reproduce the range of ON/OFF polarities observed in calcium imaging. Our modeling results are consistent with experimental results from a pharmacological block of inhibition, which increased the mixing of ON and OFF responses in VG3 dendrites[7]. Network effects of bath-applied inhibitory blockers are a serious confound of pharmacological experiments, and these results need to be confirmed by cell-type-specific manipulations. Nonetheless, we propose that the 3 to 1 ratio of inhibitory to excitatory synapses on VG3s is directly linked to the VG3's ability to transmit functionally distinct signals within its arbor.

### Plexus vs. cells as a functional unit

Mixed input/output dendrites allow one neuron to process and transmit many semi-autonomous information streams in a limited space. This arrangement has the clearest benefits in tissue with severe space limitations and where information is organized on a fine spatial scale. In vertebrates, these conditions are most common in primary sensory systems. The vertebrate retina is constrained by the number of cell bodies it can stack in front of photoreceptors (as evidenced by the contortions of the fovea). It also maintains the micrometer-scale organization of photoreceptor space using the parallel projection and stratification of bipolar cells[53]. The mixed input/output arbors of VG3s take advantage of this organization and deliver different signals to different RGC types by virtue of where their arbors overlap in the IPL.

One consequence of VG3s operating as a plexus is that the morphology of each cell is likely to be less important than that of cells like RGCs. In a mosaic of RGCs, the regularity of representation across visual space depends on each RGC of a type, forming dendritic arbors of the same size, shape, and stratification depth. A plexus of irregularly shaped VG3s (such as cell #5) could homogeneously process visual space if they fill the space between the ChAT bands evenly across the breadth of the retina. Developmental studies of differences in VG3 arborization in different mouse strains support this idea. The shape of VG3s arbors is relatively indifferent to variations in VG3 cell density. Instead, the density of VG3 dendrites tracks with the density of available bipolar cell partners[20]. A developmental stabilization of VG3 dendrites by bipolar cell inputs might also explain the tendency we observed of bipolar inputs being enriched at the terminals of VG3 dendrites.

## Methods

### Statistics and reproducibility

All the anatomical data in this study comes from a single mouse retina. Variability between animals, ages, strains, and sex are not addressed by multiple sampling. When we report error bars or confidence intervals these statistics are based on the number of cells or synapses in one piece of tissue. The population estimate we report, therefore, reflects our degree of confidence that we have measured this piece of tissue accurately and is not an estimate of the population mean for all mice or even all male C57 bk6 mice.

While data from a single piece of tissue cannot address individual variation, it can reveal patterns and mechanisms that were not previously known. When we see that our six VG3s each exhibit a strong preference for type 3a bipolar cell inputs, we are confident that this specificity is a behavior that VG3s are capable of. The VG3s of other mice may vary in their preferences, but strong individual variation in cell type preferences would be unusual for mouse retinal tissue. When

we estimate the functional length constant and model the role of inhibitory synapses, these estimates reflect the relationship between arbor morphology, synapse distributions, and neurite response properties for a single piece of tissue. Our model is, therefore, of how a single piece of tissue can work. Given the anatomical and functional consistencies between our tissue and previously published studies, we expect that our model of VG3 signal integration is roughly applicable to other mice.

### Animals

We crossed Cre-dependent GCaMP6f mice (Ai148, JAX #030328) to mice expressing Cre recombinase from locus encoding VGLUT3 (Slc17a8-IRES2-Cre-D, Jax # 028534) without disrupting endogenous VGLUT3 expression. Mice were housed in a 12 h light/dark cycle and fed ad libitum. For correlated light and electron microscopy (CLEM), a 67-day-old male mouse was dark-adapted overnight (>8 h), deeply anesthetized with $CO_2$, killed by cervical dislocation, and enucleated. All procedures in this study were approved by the Institutional Animal Care and Use Committee of Washington University School of Medicine (Protocol # 23-0116) and complied with the National Institutes of Health Guide for the Care and Use of Laboratory Animals.

The use of a male mouse was made by selecting the best tissue quality from a mixed-sex pool of experimental animals. Sex was not considered an important biological variable for two reasons. First, the retinal looming response and input-output organization of VG3s is preserved across species and unlikely to dramatically vary between sexes. Second, claims made based on our single-tissue study are only relevant as rough estimates of large effects.

### In vivo electroporation

We labeled individual VG3 amacrine cells in VG3-Cre/Ai148 transgenic mice with tdTomato (i.e., a spectrally separable activity-independent fluorescent reporter) by in vivo electroporation (Supplementary Fig. 3). To this end, injected 200 nL of pAAV-CAG-Flex-tdTomato (1.3 μg/ μl) into the subretinal space in the ventral half of the eye of P0-P1 VG3-Cre/Ai148 via a Nanoject II system (Drummond). We then delivered five 80 V square pulses of 50 ms duration using an ECM830 (BTX Harvard Apparatus) via tweezer electrodes to electroporate VG3 cells.

### Live tissue preparation

Retinas were isolated under infrared illumination in mouse artificial cerebrospinal fluid buffered with HEPES (mACSF$_{HEPES}$ for immuno-histochemistry) or sodium bicarbonate (mACSF$_{NaHCO3}$ for two-photon imaging). mACSF$_{HEPES}$ contained (in mM): 119 NaCl, 2.5 KCl, 2.5 CaCl$_2$, 1.3 MgCl$_2$, 1 NaH$_2$PO$_4$, 11 glucose, and 20 HEPES (pH adjusted to 7.37 with NaOH). mACSF$_{NaHCO3}$ contained (in mM) 125 NaCl, 2.5 KCl, 1 MgCl$_2$, 1.25 NaH$_2$PO$_4$, 2 CaCl$_2$, 20 glucose, 26 NaHCO$_3$, and 0.5 L-Glutamine equilibrated with 95% O$_2$/5% CO$_2$. Isolated retinas were flat mounted on black membrane disks (HABGO1300, MilliporeSigma, Burlington, MA, for immunohistochemistry) or transparent membrane discs (Anodisc 13, Whatman, Maidstone, UK, for two-photon imaging).

### Visual stimuli

Visual stimuli were designed in MATLAB (The Mathworks, Natick, MA) and presented via the Cogent Graphics toolbox (John Romaya, Laboratory of Neurobiology at the Wellcome Department of Imaging Neuroscience, University College London, UK). The stimuli were displayed using a UV E4500 MKII PLUS II projector (EKB Technologies, Bat-Yam, Israel), backlit by a 385 nm LED. Stimuli were focused onto the photoreceptors of the ventral retina through the substage condenser of an upright two-photon microscope (Scientifica, Uckfield, UK). All stimuli were centered on the two-photon scan field and had an average intensity of ~1600 S-opsin isomerizations/S cone/s, calibrated using a photometer (UDT Instruments S471, 268R), a

spectrometer (StellarNet, BLACK Comet), and the sensitivity profile of S-cones[54]. To determine response polarities in the receptive fields of the VG3-AC plexus, we presented bright vertical bars (height: 60–80 μm, width: 50 μm) at different locations (intervals: 25 μm, range: 800 μm) along the horizontal axis of a rectangular imaging area (height: 13 μm, width: 100 μm). Each bar was presented for 1.5 s, with a 1.5 s interval between presentations. Bars were presented in pseudorandom order.

## Two-photon imaging

Images were acquired on a custom-built upright two-photon microscope (Scientifica) controlled by the Scanimage r3.8 MATLAB toolbox. Images were recorded via a DAQ NI PCI6110 data acquisition board (National Instruments, Austin, TX). GCaMP6f was excited using a Mai-Tai laser (Spectra-Physics, Santa Clara, CA) tuned to 930 nm, and the emitted fluorescence was gathered via a 60 × 1.0 NA water immersion objective (Olympus) and filtered through consecutive 450 nm long-pass (Thorlabs, Newton, NJ) and 513–528 nm band-pass filters (Chroma, Bellows Falls, VT). This blocked the visual stimulus light (peak: 385 nm) from reaching the PMT. The laser intensity at the sample was kept at <6 mW. Images were acquired at 9.5 frames per second with a pixel density of 7.88 pixels/μm². In CLEM, we sampled twelve sites, six imaging depths in two XY positions in the ventral retina, by two-photon imaging. Each scan field was 100 × 13 μm in size. Imaging depths were measured relative to the boundaries between the IPL and the inner nuclear layers, referred to as 0% IPL depth, and between the IPL and the ganglion cell layer, defined as 100% IPL depth. These boundaries were detected in transmitted light images acquired on a temperature-compensated Si avalanche photodetector (Thorlabs, Newton, NJ), measuring the transmitted laser intensity to reveal the tissue structure. Images at different IPL depths were acquired in a pseudorandom sequence. Each functional scan started with a 30 s period without visual stimulation to permit retinal adaptation to the laser light. The total duration of functional imaging for all twelve sites was ~50 min. Retinas were continuously perfused at ~6 mL/min with warm (33 °C) mACSF$_{NaHCO3}$ equilibrated with 95% O$_2$/5% CO$_2$.

## Image processing

Images in a time series were registered to the middle frame of the transmitted light channel using rigid transformation functions built into MATLAB. Identical transforms were applied to the transmitted light and fluorescence emission channel. Registered images were median filtered (2D 3 × 3-pixel kernel). Visual stimuli and imaging time series were temporally aligned by stimulus light detection in temperature-compensated Si avalanche photodetector (Thorlabs, Newton, NJ) recorded via Scanimage r3.8.

We identified functional processing in VG3-AC dendrites with serial clustering segmentation approach adapted from our previous study[8]. After removing non-responsive pixels, this method applies a functional clustering algorithm to different image features in series, consolidating the outcome into functional regions of interest (ROIs). The algorithm reduces the dimensionality of response features through principal components analysis and integrates a connectivity matrix via a K-nearest-neighbor approach to accomplish community detection clustering. Normalized signals from remaining pixels were fed back into the iterative algorithm to generate connected and functionally homogeneous pixel clusters forming ROIs. ROIs were iteratively merged and scrutinized until a stable solution was derived, typically within 15 iterations. ROIs of <5 pixels were discarded.

The estimation of the polarity of each ROI was determined by utilizing the median responses of four repeated flashes of bars in each x-position. The polarity index (PI) of each ROI was calculated as the average difference between ON and OFF responses for each flash bar position divided by the sum of the responses:

$$PI = \frac{\sum_i^n Response_{ON} - Response_{OFF}}{\sum_i^n (Response_{ON} + Response_{OFF})}$$

The ON and OFF responses are defined as the calcium signal that occurred between 0.05 s and 1.2 s following the onset and offset of the flash bar, respectively.

## Correlated light and 3DEM

EM reconstruction was targeted to the functionally characterized dendrites of VG3 neurons by comparing optical maps of the retinas to low-resolution EM reconstructions of the same tissue (approach described in detail in ref. 18). Transmitted light images and 2-photon structural images of vGlut3 expression (Fig. 1a) linked the functionally imaged regions to tissue landmarks (VG3 GCamp6 expression, blood vessels, retina edges, optic nerve). After live imaging, tissue was fixed in 2.5% paraformaldehyde and 1% glutaraldehyde and mapped using confocal microscopy. Confocal maps were acquired over a range of magnifications using GCamp6 fluorescence and tissue-intrinsic signals (aldehyde autofluorescence and reflected light, Fig. 2a). These maps provided anatomical context surrounding the functional ROI. In particular, the position of cell nuclei revealed in autofluorescence imaging was matched to the position of nuclei in medium-resolution EM image volumes (Fig. 2a–c). After optical mapping, tissue was stained, embedded, and cut into 1000 40 nm thick sections for EM imaging.

Three-dimensional volumes of the ultrathin sections were acquired at 600 nm XY-resolution so that blood vessels could be used to identify the functionally characterized region of interest. That region was then imaged at 20 nm XY-resolution to identify the functionally characterized cells (Fig. 2b). We identified the functionally characterized VG3s by their soma position and we reconstructed their arbors with manual annotation using VAST[19] (https://lichtman.rc.fas.harvard.edu/vast/). VG3 identification was aided by the observation that they have an unusual, though not unique, pattern of diffuse heterochromatin in the nucleus (Fig. 2c). With the functionally characterized VG3s identified in the medium-resolution EM volume, we then acquired a 4 nm XY-resolution volume (~100 μm × 100 μm × 40 μm deep) that encompassed the functionally characterized neuropil (Fig. 2d, e). Our identification of VG3 neurons was later confirmed when EM reconstructions of arbor morphology were compared to 3D two-photon and confocal reconstructions of the same cells.

## Circuit analysis and visualization

Cells and synapses were traced manually in VAST with volume painting that approximated the size and shape of neurite cross sections, but that were not intended to be pixel-perfect. Segmentation labels included cell and synapse identifiers such as ribbon synapses being labeled "cid4 rib (cid12 cid14)" for cell 4 forming a ribbon that innervates cell 12 and cell 14. Segmentation tags were parsed by CellNav to create lists of all cells and synapses in the volume as well as surface mesh files for each cell and segmentation object. CellNav then allowed for browsing of the reconstruction by cell type, cell ID, or connectivity. Additional information on cells such as which cell IDs should be merged together (alias list) and cell type identification were entered into a Google spreadsheet that was then read by CellNav. The ability to easily group cells by type or manually entered lists was important for efficient browsing, rendering, and analysis of the circuit.

Cells were skeletonized using a custom shortest path algorithm[37]. Errors in skeletonization generated by segmentation merges were corrected in the CellNav skeleton editor. Distance matrices for (1) the path between each node on the skeleton and (2) each synapse on the skeleton were precomputed for downstream analysis of topography and signal flow.

### To view or reanalyze the VG3 network reconstruction

Running CellNav and analysis will require Matlab with App Designer. Download CellNav_0.8 and the IxQ_CellNavLibrary. Execute runCellNav.m from the main directory of CellNav. Select the IxQ_CellNavLibrary folder. Use the 'Select Volume' dropdown to choose the "HighRes" volume. Most analysis scripts will require first executing "loadCellSMs.m" to load the skeleton files before the individual analysis scripts can be executed. The image volume can be browsed by selecting cells individually or by cell type in the" Select Cells" window. Synapses between cells or groups of cells can be visualized using the "Synapses" tab at the bottom. See CellNav_Help.pdf for additional instructions.

### Monte Carlo analysis of cell type connection probability

The depth distribution profiles for bipolar cell and RGC cell types were downloaded from EyeWire Museum. We calculated a depth correction factor to fit the stratification of our cell types to the IPL depths reported in the EyeWire dataset. The total arbor in the IPL was normalized to be equal for all RGC types to account for the undercounting of large field RGCs. We then calculated a depth overlap curve between the skeletonized arbor of our VG3 cells and each bipolar cell and RGC type in the EyeWire Museum (MonteRgcVGC_general.m). For each of 10,000 repetitions, we redistributed the classified VG3-to-RGC synapses (321) or bipolar cell-to-RGC synapses (319) in our dataset with probabilities defined by the depth overlap between VG3s and the bipolar cell or RGC types. Confidence intervals presented reflect the range of ranked results, centered at the median, that are included within the confidence interval fraction.

### Testing specificity

Testing the ON/OFF specificity of synapse distributions. We tested whether there was specificity to the ON/OFF distribution of synapses beyond stratification. We used a Monte Carlo model in which synapses of VG3s were redistributed on the VG3 arbor while maintaining their depth in the IPL ($\pm 0.5\,\mu m$). We then tested how the 1000 such randomizations changed the predicted polarity of the dendrite that different types of synapses would find themselves in using a $16\,\mu m$ length constant. To quantify specificity, we measured how much the predicted polarity of synapses shifted towards the mean of the cell when bipolar input positions were randomized.

### Estimating length constant

We found that the median difference between ROI polarities less than $3\,\mu m$ apart was 0.214 (242 correlation points, polarity measure ranges from −1 to 1). For estimating a reasonable length constant, predicted the polarity of each point on our VG3 skeletonization using an exponential decay model and a range of length constants. We assigned a value of one to each bipolar cell input. We then calculated the distance between each bipolar cell input and each tested correlation point. For each tested length constant ($lc = 0$–$150\,\mu m$), we calculated the influence (W) of each bipolar cell on each correlation point given the distance (d) between them and exponential decay with the given length constant ($W = e^{-d/lc}$). We summed the ON influences and OFF influences for each correlation point and determined the polarity as $(ON − OFF)/(ON + OFF)$.

To determine the length constant that best matched our anatomical data to previously published depth polarities of VG3s[8] we used the following target polarities: depth as a fraction of IPL [0 .21 .29 .37 .44 .51 .60], mean polarity [−.18 −.44 −.45 −.325 0.02 0.225 0.23] ("makePOI.m"). To find the anatomic prediction of polarity for each depth, we averaged nodes $0.25\,\mu m$ above and below the target depths. We found the best-fit length constant by minimizing the summed differences between the predicted polarity and the published polarities (Supplementary Fig. 6b).

To find the best length constant for matching our anatomic prediction to functional recordings of the same cells (Supplementary Fig. 6a), we first tested the noisiness of the functional estimates of polarity. We compared the difference in polarity estimates for pairs of EM identified locations with the distances between them on the VG3 arbor ("analyzeROInoiseByLength.m"). Our assumption is that ROIs less than $3\,\mu m$ apart should have essentially the same ON/OFF polarity. We found that the median difference between ROI polarities less than $3\,\mu m$ apart was 0.214 (242 correlation points, polarity measure ranges from −1 to 1). We found that one of our frames was responsible for introducing most of the errors and so we eliminated that frame from future analysis (reduced to 202 comparisons with a median polarity difference of 0.156). We also found that we could reduce the polarity differences of nearby ROIs difference to 0.122 by eliminating ROIs with significant inter-trial variations in polarity (reduced to 54 correlation points). A polarity difference of about 0.12 can then also be considered a ceiling for the accuracy with which we might be expected to match functional measurements of polarity with anatomical estimates.

In running the model comparing anatomic predictions to recordings of the same cell, we added a noise parameter to the anatomic estimate. The noise parameter was a range (0:0.001:0.005) that was added to the influence between each bipolar cell and each correlation point. This constant was intended to compensate for the fact that random pixel noise in functional recordings would have the tendency to pull the polarity estimates (−1 to 1) towards 0. We found the best fit between the functional recordings and the anatomic predictions by summing the difference in the two polarities across all correlation points (jmFindLengthConstant15.m).

To estimate a length constant without reference to polarity we compared the correlation coefficient of functional responses of pairs of points identified in light and EM with the similarity of their bipolar cell input (Supplementary Fig. 6c) ("jmFindLengthConstantAutoCorr05.m"). Functional correlations were measured for pairs of correlation points that were identified in the same functional imaging plane. Bipolar cell influences on these points were calculated as described above. Their anatomic correlation coefficient was calculated by finding the correlation coefficient of the lists of bipolar cell influence for each point. The best fit was calculated as the peak correlation coefficient between when matching the correlations of anatomic predicted correlations using different length constants and the functional correlations.

### Modeling the VG3 plexus with NEURON

Electrotonic modeling was performed using the t2n[55] Matlab (Mathworks) package for running models in NEURON[33]. Six VG3 arbors were converted into swc skeleton format with ~0.1 μm internode distance. The swc skeletons were converted to NEURON hoc format using "load_tree.m" and "t2n_writeTrees.m" within "jmRunNEURON_VG3.m". Membrane capacitance (cM = 1xx) and axial resistance (Ra = 100) were set as standard values for NEURON modeling. The resting potential (−50 mV), excitatory reversal potential (0 mV), and inhibitory reversal potential (−60 mV) were set based on previous recordings of VG3s[5]. Models were run at 33 °C to match the temperature at which calcium activity is recorded. Synapses were simulated using Exp2Syn point processes with a rising tau of 1 ms and a falling tau of 1 ms for excitatory synapses and 3 ms for inhibitory synapses. Each experiment consisted of activating either OFF bipolar cell inputs and all inhibitory synapses or ON bipolar cell inputs and all inhibitory synapses. A minimum node diameter of 0.1 μm was enforced to prevent possible segmentation inconsistencies from disrupting the model. Maximum membrane voltages were recorded for all skeleton nodes.

Four parameters were varied to find the best fits between the VG3 models and the recorded calcium polarities: passive membrane conductance (Gm), excitatory synapse conductance (Ge), inhibitory synapse conductance (Gi), and a scaling factor to reduce inhibition during ON vs OFF stimulation (OnScale). Synapse conductance was applied as the weight of the connection to the artificial neurons driving the synapses. The conductance of inhibitory synapses was multiplied

by the OnScale parameter during ON excitation experiments. The range of values tested for each parameter was determined manually by comparing combinations of parameters that bounded physiologically plausible values (Gm 10^[−7: −1] nA, Ge 10^[−4: −1] nA, Gi 10^[−Inf: −2] nA, OnScale [0.1: 1.25]). Experiments were focused around emerging local minima in the model/recording comparisons. A total of 296 experimental combinations were combined by "jmCollectNEUR-ON_VG3.m" for consideration.

NEURON results were analyzed and displayed with "jmShow-NEURON_VG3.m". Model best fits were determined by comparing the mean error of the OFF bias between the model and the calcium recordings. To reduce the influence of oversampled polarities, the weight of grouped functional rois was scaled by the density of sampling at that polarity. The performance of the models was further compared by finding the correlation coefficient between the predicted polarities and the calcium polarities and by finding the ON/OFF match. ON/OFF match was defined as the percent of rois with the correct binarized bias (ON or OFF).

To test the effect of neurite diameter variation on the polarity predictions, the diameter of all skeleton nodes was set to 0.5 μm, the approximate median diameter of VG3 neurons.

### Bipolar cell to RGC through VG3 influence matrix

The influence of each bipolar cell type on each RGC type via the VG3 plexus was calculated by first finding the estimated influence of each bipolar cell to VG3 synapse on each VG3 to RGC synapse using exponential decay and a length constant of 16. The bipolar cell influences were then summed according to bipolar cell type. All VG3 to RGC synapses were then summed according to RGC type ("calculateBipToRGCthroughAMC.m").

### Monte Carlo prediction of VG3 to RGC polarity

To determine if stratification depth was sufficient to predict the signal polarity that VG3s deliver to each RGC type, we ran a Monte Carlo distribution where synapses between VG3s and RGCs were redistributed ("predictPolarityBasedOnArborOverlap.m"). We used the overlap between our VG3 arbors and EyeWire Museum RGCs to generate overlap with depth curves for VG3s and each RGC type. We then redistributed the synapses between VG3s and each RGC type across the VG3 arbor with the probability of the new synapses occurring at any given depth being determined by the VG3-RGC depth overlap profile. The redistribution was repeated 10,000 times. For each iteration, we obtained the ON and OFF bipolar cell influence onto each new VG3 to RGC synapse (length constant 16 μm). We summed these influences for each RGC type and found the polarity as (ON − OFF)/(ON + OFF). Quartile whiskers represent the 0.001−0.25, 0.25−0.5, 0.5−0.75, and 0.75−0.999 intervals of the sorted results.

We calculated a confidence interval for the range of polarity differences between synapses innervating ON and OFF RGCs using a bootstrap method ("bootDifCI.m"). The polarity from each group was resampled 10,000 times with replacement. We found the confidence interval that bounded 95% of the resulting differences in means. We then subtracted these bounds from two times the observed difference in means to find the final confidence interval. The interval generated by this bootstrap method (0.39−0.60) was close to the parametric estimate of the 95% confidence interval using the standard error of the differences ("standardDifCI.m")(0.38−0.56).

### Triad motif analysis

Triads were identified either during manual reconstruction or by searching for appositions between bipolar cells and RGCs that were synaptically connected to the VG3 plexus. The apposition detector (in CellNav analysis) was set to find all points where the meshes of the potential synaptic partners came withing 0.1 μm of one another. The resulting list of apposition points was then manually spot checked for

synapses, resulting in annotation of 297 bipolar to RGC synapses. For four individual RGCs representing types frequently targeted by VG3 synaptic output, input synapses were manually annotated across their dendritic arbors, resulting in 336 additional bipolar to RGC synapse annotations. For these RGCs, we then determined the polarity of bipolar cells providing inputs, whether those bipolar cells also innervated the VG3 plexus, the relative ratios of bipolar, amacrine, and VG3 input, and the distributions of these inputs on the RGC arbor branches.

### Reporting summary

Further information on research design is available in the Nature Portfolio Reporting Summary linked to this article.

## Data availability

All optical images, EM images, and segmentation will be made available upon request. Links to data and annotations are posted to https://sites.wustl.edu/morganlab/vg3-data-code-and-more/

## Code availability

All code used for data acquisition and analysis (besides ScalableMinds alignment code) can be found at https://github.com/MorganLabShare/VG3_2023 (https://doi.org/10.5281/zenodo.10685346). Code and instructions can also be found at https://sites.wustl.edu/morganlab/vg3-data-code-and-more/.

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

## Acknowledgements

Thanks go to the Washington University Center for Cellular Imaging and staff where the EM was performed and to ScalableMinds that aligned the EM dataset. Thanks also to Noah Bastola, Randi Anderson, and Pablo Kerschensteiner for tracing. Thanks to Richard Schaleck and the Jeff Lichtman lab for providing carbon-coated Kapton collection tape. This work was supported by an unrestricted grant to the Department of Ophthalmology and Visual Sciences from Research to Prevent Blindness, by a Research to Prevent Blindness Career Development Award (J.L.M.), and by the NIH (EYE030623 to D.K. and J.L.M., EYO29313 to J.L.M., EYO26978, EYO23341, and EYO27411 to D.K.).

## Author contributions

J.L.M., D.K., K.F., and J.H. designed the experiment. J.H. performed the live imaging. K.V. processed tissue for EM. K.F. performed the EM data collection. J.L.M., K.F., K.V., and L.M. segmented the EM volume. J.L.M. and K.F. analyzed the circuit reconstruction. J.L.M., K.F., D.K., and J.H. wrote the manuscript.

## Competing interests

The authors declare no competing interests.
