## [Peer Review File · Nature Communications]

REVIEWER COMMENTS

Reviewer #1 (Remarks to the Author):

Review by Greg Schwartz

The manuscript by Friedrichsen and colleagues combines functional calcium imaging with 3D electron microscopy (EM) to investigate subcellular processing in VGluT3-expressing amacrine cells (VG3) in the mouse retina.

The EM annotation in this study and its reporting and statistics are exceptionally well done. While most aspects of the results are not necessarily “surprising,” the authors convincingly show the novel result of wiring specificity from BC3a bipolar cells to VG3 cells. These aspects make it an important contribution to the field.

The conclusion that VG3 cells signal locally within their neurites rather than integrating signals was already established in previous papers from the Kerschensteiner lab and others, but this manuscript takes it a step further by measuring the local connectivity matrix of inputs and outputs to VG3 cells to speculate about the details of that local processing. I use the word “speculate” because one can’t conclusively determine much about the nature of subcellular processing from anatomical measurements alone, particularly with the limitation that the authors were not able to distinguish excitatory from inhibitory output synapses, and VG3 cells contain both. For the most part, however, the manuscript is written in a way that does not overreach or overinterpret. One of the take-home messages for me is that detailed measurements and analyses of the locations of input and output synapses are necessary, but not sufficient, for understanding subcellular information processing in neurons.

Major suggestions

1. The influence of the RF surround (or, more generally, inhibition) on the calcium responses of VG3 cells is worth additional consideration as it relates to the ON-OFF index calculation that becomes important in several parts of the manuscript. The retina was stimulated with long vertical bars that extended well outside the imaging window. Thus, photoreceptors upstream of the VG3 cells were exposed to two different contrasts depending on their location. Those within the imaging window received a contrast of $(\text{stim_intensity} + \text{laser_intensity}) / \text{laser_intensity}$, while those outside the window received a contrast of $(\text{stim_intensity} + \text{nominal_zero}) / \text{nominal_zero}$. Thus, the contrast outside the imaging window was higher than that inside, and it might have been by several orders of magnitude. Depending on unknown details of the upstream circuits, this could mean that the calcium responses in the VG3 dendrites were actually dominated by inputs originating from outside the scan field instead of from the bipolar cells

within the scan field. The authors could get a sense of this by repeating the imaging experiments with bars or spots of different sizes (and ideally with imaging windows of different sizes) and seeing if this affects the ON-OFF index in each stratum. Some of this data appears to be in the eLife paper on VG3 cells from the Kerschensteiner lab. It might just have to be reanalyzed slightly for this purpose. Of course, I don't expect them to repeat the EM parts on the same samples, but this functional imaging experiment should be manageable and would provide some helpful context in which to understand whether ON-OFF index is a fixed property of a VG3 stratum or whether it depends on the size of the stimulus.

2. As mentioned above, the manuscript contains some speculation about the function of VG3 neurites and how they might carry "processed" signals to RGCs from the same bipolar cells that are also providing direct input to the RGCs. The authors mention that even if it is not input from exactly the same set of bipolar cells, it is typically from ones of the same ON or OFF polarity. Much of the last three sections of Discussion is about this topic broadly. The problem is that the authors were unable to distinguish between glutamatergic and glycinergic output synapses from VG3 cells. Without this information, there is an unknown sign inversion, so it becomes hard to say much of anything concrete about function. With both output signs possible, anatomically OFF-only RGCs could receive OFF excitation and/or OFF inhibition from VG3 ACs, and ON-only RGCs could receive ON excitation and/or ON inhibition (but also maybe OFF signals since responses in ON-stratifying VG3 dendrites are mixed as shown in Fig. 1). The motif of the same bipolar cells providing input to an RGC directly and indirectly through the VG3 is interesting, and the relative ON-OFF isolation in VG3 suggests that it does not support much crossover inhibition, at least from the ON pathway to the OFF pathway (which is a known function of the A2 AC). But other than that, I'm not sure there is much that can be known from this data. I completely understand that the neurotransmitter of the synapse could not be determined from the EM data, but the authors might want to scale back their speculations about function given this. Future experiments of this type could try to identify synapse type either with an antibody in fluorescence microscopy before EM or in immuno-gold during EM.

3. The functional length constant analysis is a really nice aspect of this study, but it seems incomplete to me.

A. While I appreciate that the authors used three different methods to compute the length constant of the calcium signals, the text explaining each method is pretty dense. A visual example of each method in the figure would be helpful. Also, as far as I understand (which might not be totally correct), the methods are not particularly independent. Method 1 uses data from a previous study, but it's essentially the same experiment as for the data in the other two methods. Methods 2 and 3 use the same raw calcium traces but group the pixels differently in space and in time (but of course, they are highly correlated in both space and time). This is not really a criticism of the methods themselves, but some more context about how they are similar or different and interdependent would help the reader.

B. More importantly, these are all functional measures of the length constant of calcium signals; none are estimates of either the electrical length constant of the neurites or the length constant of the synaptic output (glutamate or glycine release). The authors acknowledge the later issue to some extent, but they kind of sweep it under the rug. Yes, calcium is related to synaptic output, but this relationship can be highly local in space (more so than is captured by GCaMP6f), and it is almost always highly nonlinear, with a hill coefficient of 4 or so. Given these uncertainties, which are even more uncertain in a mixed population of glutamate and glycine synapses, I would argue that calcium signals are a very indirect measure of synaptic output. This indirectness should be reflected in the way the authors write about this relationship.

C. The question of the assumed electrical length constant vs. calcium is potentially more interesting and addressable with an additional modeling component. As the authors point out, they do not know the full complement of ion channels on VG3 neurites, so they can't make an accurate active model. However, they can make a passive cable model of the VG3 arbor. All they need for this model is the membrane resistivity and the axial resistance (which is typically assumed to be proportional to neurite diameter with a known proportionality constant). Patch-clamp recordings from previous studies should constrain the membrane resistivity based on the total resistance of a VG3 cell, but even in the absence of this measurement, the authors could test a realistic range of values based on measurements from other amacrine cells or RGCs. Since the authors have 3D reconstructions of the neurites, they are in an excellent position to build such a passive model. I suspect that the electrotonic length constant in such a model within any reasonable range of the parameter values would still be dramatically larger than the value of $\sim 16 \mu\text{m}$ that they arrived at in the functional measurements. If I am wrong and some aspect of VG3 neuritic morphology helps to make it especially electrically isolated, that is a fascinating result that deserves more analysis. If I am right and the passive electrical properties of the VG3 cable model are incompatible with the functional data, then this itself is an important result worthy of reporting. This result would suggest that active properties of the VG3 neurites and/or a highly nonlinear relationship between voltage and calcium influx lead to greater functional isolation in the VG3 than one might expect.

Minor suggestion

1. Line 202: alpha character not printed correctly

Reviewer #2 (Remarks to the Author):

This study examines the link between the functional property of VG3 dendrites and their spatial organization of synaptic inputs and outputs. The authors first did subcellular two-photon calcium imaging to determine the polarity of the light responses across dendritic locations in the On and Off strata in the inner plexiform layer (IPL), and then performed connectomic reconstruction of the imaged regions. They analyzed the spatial distributions of VG3 input and output synapses across the IPL depth, and evaluated how this wiring pattern contributes to the functional properties of dendritic subregions. This is a powerful approach that fills the missing functional information in many conventional connectomic analyses. This study provides valuable information about the synaptic basis of VG3 circuit function and uncovers several intriguing features of VG3 synaptic organization, such as the distinct patterns of excitatory and inhibitory synapse distribution, biased innervation by bipolar cell type 3a, and shared bipolar cell inputs between VG3 and its postsynaptic RGC targets. It's an impressive effort and sets an example of combining functional characterization and connectomic analysis to uncover synaptic mechanisms of visual processing. I only have some minor suggestions to improve the clarity of the manuscript.

Given that ~ 74% of the synaptic inputs onto the VG3 cell are from amacrine cells, it leaves the reader wondering what these inhibitory inputs come from (i.e. from what amacrine cell types, and how they shape the dendritic responses. More information about these points will further strengthen this study.

Lines 219-220: " we randomly pruned synapses so that synapses between pairs of connected cells were at least 10 micron from their nearest neighbor." Can the authors provide more information about the method of random pruning and the rationale behind it? Why and how does this method "reduce biases introduced by observing only a few examples of each post synaptic cell type"?

Lines 225-231: Given the EM dataset cannot distinguish between glutamatergic and glycinergic outputs of the VG3, the percentages don't necessarily reflect the VG3's contribution to the excitatory inputs. Along the same lines, lines 313-315, for the example of monostратified On cells, it is also possible that the VG3 synapses biased towards Off responses are inhibitory. This complication might contribute to the difference between the bipolar cell population directly connected to the RGC and the mix that is predicted to influence it through the VG3s (lines 346-348).

Lines 351-355: I found these sentences difficult to understand.

Fig. 6 "d" label is missing

Reviewer #3 (Remarks to the Author):

The article by Friedrichsen et al entitled "Subcellular pathways through VG3 amacrine cells provide regionally tuned object-motion-sensitive signals in mouse retina" combined functional imaging and connectomic reconstruction and revealed synaptic input and output profiles in VGluT3-expressing amacrine cells (VG3s). The model evaluations of the mapped connectivity suggested a subcellular wiring diagram of input and output connectivity through a VG3 plexus.

As I read the introduction part, I understood that an appeal of this manuscript is the combination of physiological imaging and connectomics. However, individual results are not correlated well to present "novel" insight into retinal processing. Indeed, most EM analyses (Figure 2-7) were evaluated by modeling and independent of the imaging results. For example, an important correlation between physiology and anatomy could be to examine how the identified input synapses (spatial profiles, cell types) are integrated and establish the dendritic activity required for the looming sensitivity. While the authors discovered a prominent input from type 3a bipolar cells out of other cell types, how such dominated excitatory inputs can be correlated with actual VG3 responses was not examined. The authors showed an analysis to correlate characterized physiology (ON and OFF response property) and the mapped synapses (Figure 1). However, this response polarity shift is not a novel result (e.g. Chen et al., PNAS, 2017; Hsiang et al., eLife, 2017).

Also, the authors suggested a function of VG3 plexus, rather than a single cell unit, to provide local object-motion-sensitive signals with postsynaptic retinal ganglion cells (RGCs). However, the contributions of VG3 to RGC response were not substantial (Figure 6f). Indeed, previous studies have shown that only small subsets of RGC types inherit the looming sensitivity of VG3. In that sense, the impacts of the VG3 plexus would be negligible for most RGCs innervating the plexus.

Further details to be considered are described below.

Major comments:

1) The correlation between physiological imaging and anatomical synapse mapping. The authors engaged in the anatomical analysis: the synaptic input mapping (Figure 3) and the spatial profiles (Figure 4). On the other hand, the relationship between the characterized synapses (locations, cell types) and physiology (dendritic activity of the individual VG3s) is unclear. Although the authors mapped the response polarity of the dendrites at the different IPL depths, this has already been shown by previous studies (e.g. Chen et al (PNAS, 2017); Hsiang et al (eLife, 2017)). To add new insights into dendritic computation in VG3, the authors should correlate those anatomical and physiological findings and propose how the mapped synaptic inputs form the dendritic activity. In particular, it is important to

discuss how the identified bipolar cell (and amacrine cell) inputs are correlated with the known synaptic input properties generating looming sensitivity.

2) The authors found that the density of amacrine cell inputs was highest between ON and OFF bipolar cell input zones, and suggested that the inhibitory inputs from amacrine cells could contribute to the ON and OFF segregation (line 115-121). If this is true, the dendritic excitation (Ca²⁺ signal) around the middle of VG3 dendritic arbors could be shunted and resulted in less Ca²⁺ signal, compared with the proximal (OFF response) and distal (ON response) dendrites. However, the response polarity mapping (Figure 1d; Chen et al., PNAS, 2017) showed ON-OFF responses at the middle VG3 dendritic arbors ("Depth ~0.535, 0.536" in Figure 1d left). Also, previous studies (Chen et al., PNAS, 2017; Hsiang et al., eLife, 2017) did not show evident differences in Ca²⁺ signal intensity across dendritic segments. The authors should discuss how the spatial profile of amacrine cell inputs can contribute to the observed response polarities.

3) Based on the differences in the observed synapses and Monte Carlo (Figure 4f), the authors revealed an evident synaptic input from type 3a bipolar cells, compared with other bipolar cell types. However, it is unclear whether the probabilities of synapse formations are different across bipolar cell types or whether the density of dendritic arbors of VG3s at the different IPL depths affects the number of synapses. For example, a sparser plexus of the proximal dendrites, in which type 3a bipolar cell innervates, than the distal plexus could result in distinct synapse numbers. The authors should clarify whether the dendritic morphology affects it or not.

4) VG3s innervation to RGCs. The authors found that VG3 innervates different RGC types (Figure 5). Although 5ti and 4ow types inherit a VG3 function (sensitivity to looming stimuli), other RGC types do not. The authors should discuss why the VG3 innervations are not functional in the RGC types regardless of the unignorable VG3 inputs.

5) The authors modeled bipolar cells-RGCs transmission through VG3 (Figure 6), and claimed that there were deviations between the predicted RGC response polarity and the actual response despite the strong influence of type 3a inputs (Figure 6f). However, each RGC type receives not only bipolar inputs through VG3 (bipolar-VG3-RGC), but also direct bipolar cell inputs (bipolar-RGC). If the impact of bipolar-VG3-RGC pathways is minor compared with that of bipolar-RGC pathways, this model cannot explain the RGC response well. The authors should clarify the validity of this modeling to analyze the impact of bipolar cell inputs on RGC properties.

6) The authors claimed that a VG3 plexus provides local object-motion-sensitive signals with the postsynaptic RGC types. However, this should be examined experimentally. There is no "direct" evidence of the localization of the signal (only models in Figure 6). Even if RGCs innervating the VG3 plexus receive such local VG3 inputs, it is not evident if the VG3 inputs affect the RGC responses. Indeed, the excitatory

impacts of VG3 to explain the postsynaptic RGC response were not high (Figure 6f). Therefore, to clarify the function of VG3 plexus to provide local object-motion-sensitive signals, the authors should perform Ca²⁺ imaging from the targeted RGC dendrites and examine the local dendritic activity in the looming stimuli.

Minor comments:

1) There are no explanations of "AMC" and "BPC" (Figure 2); "VGC" (Figure 6).

2) Figure 4. It is not clear why the population of bipolar cell types in individual VG3s looks very different from the population in "All cells". For example, the ratio of bc3a inputs is the highest in all cells. However, the populations in the individual cells are not high.

3) Figure 4f. To examine the physiological relevance of the less ON bipolar cell inputs, the authors should analyze the differences in intensities of ON and OFF dendritic Ca²⁺ signals.

4) Labels of individual VG3s in Figure 5b right (e.g. cid = 2) should be related to Figure 4d right (e.g. #2).

Dear Reviewers

Thank you for agreeing to review this manuscript. We have read your comments carefully and have made additions and changes to the manuscript to address your concerns. Additions include electrotonic modeling of VG3s using NEURON and additional functional imaging of VG3s. We have also revised the way we talk about the predictions of our model. We no longer say that we are predicting the signal polarity delivered to RGCs. Instead, we say that we are predicting the response polarity of the VG3 neurites that innervate RGCs. Please see the detailed responses to individual comments below.

R01Reviewer #1 (*Remarks to the Author*):

Review by Greg Schwartz

The manuscript by Friedrichsen and colleagues combines functional calcium imaging with 3D electron microscopy (EM) to investigate subcellular processing in VGluT3-expressing amacrine cells (VG3) in the mouse retina.

The EM annotation in this study and its reporting and statistics are exceptionally well done. While most aspects of the results are not necessarily “surprising,” the authors convincingly show the novel result of wiring specificity from BC3a bipolar cells to VG3 cells. These aspects make it an important contribution to the field.

The conclusion that VG3 cells signal locally within their neurites rather than integrating signals was already established in previous papers from the Kerschensteiner lab and others, but this manuscript takes it a step further by measuring the local connectivity matrix of inputs and outputs to VG3 cells to speculate about the details of that local processing. I use the word “speculate” because one can’t conclusively determine much about the nature of subcellular processing from anatomical measurements alone, particularly with the limitation that the authors were not able to distinguish excitatory from inhibitory output synapses, and VG3 cells contain both. For the most part, however, the manuscript is written in a way that does not overreach or overinterpret. One of the take-home messages for me is that detailed measurements and analyses of the locations of input and output synapses are necessary, but not sufficient, for understanding subcellular information processing in neurons.

We thank the reviewer for their careful reading, positive remarks, and constructive criticism of our manuscript. We address specific comments below.

Major suggestions

1. The influence of the RF surround (or, more generally, inhibition) on the calcium responses of VG3 cells is worth additional consideration as it relates to the ON-OFF index calculation that becomes important in several parts of the manuscript. The retina was stimulated with long vertical bars that extended well outside the imaging window.

Thus, photoreceptors upstream of the VG3 cells were exposed to two different contrasts depending on their location. Those within the imaging window received a contrast of $(stim_intensity + laser_intensity) / laser_intensity$, while those outside the window received a contrast of $(stim_intensity + nominal_zero) / nominal_zero$. Thus, the contrast outside the imaging window was higher than that inside, and it might have been by several orders of magnitude. Depending on unknown details of the upstream circuits, this could mean that the calcium responses in the VG3 dendrites were actually dominated by inputs originating from outside the scan field instead of from the bipolar cells within the scan field. The authors could get a sense of this by repeating the imaging experiments with bars or spots of different sizes (and ideally with imaging windows of different sizes) and seeing if this affects the ON-OFF index in each stratum. Some of this data appears to be in the eLife paper on VG3 cells from the Kerschensteiner lab. It might just have to be reanalyzed slightly for this purpose. Of course, I don't expect them to repeat the EM parts on the same samples, but this functional imaging experiment should be manageable and would provide some helpful context in which to understand whether ON-OFF index is a fixed property of a VG3 stratum or whether it depends on the size of the stimulus.

Thanks for raising this important point. To explore the impact of the two-photon laser scanning (and the uneven distribution of scan regions between the receptive field center and surround) on polarity indices, we acquired and analyzed data across IPL depths with scan regions differing four-fold in size (1,936 μm^2 vs. 7,744 μm^2). Compared to IPL depth and stimulus size (see below), the scan size had only a minor effect on response polarities (Supplementary Fig. 1a).

At a given IPL depth, response polarities are not a fixed property of VG3 dendrites; they depend on stimulus size. We demonstrate this by reanalyzing previously acquired data (Hsiang et al. 2017) to calculate polarity indices across depths for responses elicited by spots varying in size (diameter: 50 μm to 200 μm). The respective curves (polarity indices vs. imaging depth) shift along the y-axis with increasing stimulus size, indicating that the relative activation of ON and OFF inputs differs between stimuli of different sizes rather than the extent of their mixing, which would be expected to alter the slope (Supplementary Fig. 1b). Consistent with this interpretation differences in the receptive field sizes of ON and OFF inputs to VG3 dendrites can be observed in patch-clamp recordings from VG3s (e.g., Kim et al. 2015 reported receptive field center sizes of ON excitation: $137 \pm 15.8 \mu\text{m}$ and OFF excitation: $83.1 \pm 10.2 \mu\text{m}$, $p < 0.005$) and in the dendritic calcium transients of VG3s (ON center: $80.4 \pm 24.1 \mu\text{m}$, OFF center: $54.4 \pm 18.2 \mu\text{m}$, $p < 0.001$).

Finally, we reanalyzed data from Kim et al. (2015) and Hsiang et al. (2017) to show that polarity indices of somatic responses of VG3s vary in patch-clamp recordings and two-photon calcium imaging (Supplementary Fig. 1c). We hope that inclusion of the new supplementary figure (Supplementary Fig. 1) clarifies our interpretation of response polarity indices.

2. As mentioned above, the manuscript contains some speculation about the function of VG3 neurites and how they might carry “processed” signals to RGCs from the same bipolar cells that are also providing direct input to the RGCs. The authors mention that even if it is not input from exactly the same set of bipolar cells, it is typically from ones of the same ON or OFF polarity. Much of the last three sections of Discussion is about this topic broadly. The problem is that the authors were unable to distinguish between glutamatergic and glycinergic output synapses from VG3 cells. Without this information, there is an unknown sign inversion, so it becomes hard to say much of anything concrete about function. With both output signs possible, anatomically OFF-only RGCs could receive OFF excitation and/or OFF inhibition from VG3 ACs, and ON-only RGCs could receive ON excitation and/or ON inhibition (but also maybe OFF signals since responses in ON-stratifying VG3 dendrites are mixed as shown in Fig. 1). The motif of the same bipolar cells providing input to an RGC directly and indirectly through the VG3 is interesting, and the relative ON-OFF isolation in VG3 suggests that it does not support much crossover inhibition, at least from the ON pathway to the OFF pathway (which is a known function of the A2 AC). But other than that, I’m not sure there is much that can be known from this data. I completely understand that the neurotransmitter of the synapse could not be determined from the EM data, but the authors might want to scale back their speculations about function given this. Future experiments of this type could try to identify synapse type either with an antibody in fluorescence microscopy before EM or in immuno-gold during EM.

We agree with these points and have adapted the text in two ways. In the results section, we have changed the way we talk about the results of our modeling so that our predictions are of the calcium polarity of neurites that are synaptically connected to different RGC types. We have removed references to predicting the synaptic drive onto RGCs. We think this change is both more clear and more accurate. In the discussion we now have a section that directly addresses how the reviewers above issues apply to the question of translating predictions of presynaptic calcium polarity to transmitter release from those neurons onto RGCs.

3. The functional length constant analysis is a really nice aspect of this study, but it seems incomplete to me.

A. While I appreciate that the authors used three different methods to compute the length constant of the calcium signals, the text explaining each method is pretty dense. A visual example of each method in the figure would be helpful. Also, as far as I understand (which might not be totally correct), the methods are not particularly independent. Method 1 uses data from a previous study, but it’s essentially the same experiment as for the data in the other two methods. Methods 2 and 3 use the same raw calcium traces but group the pixels differently in space and in time (but of course, they are highly correlated in both space and time). This is not really a criticism of the methods themselves, but some more context about how they are similar or different and interdependent would help the reader.

We have revised the description of the models used for estimating the length constant to improve clarity. We also provided a supplementary figure to help readers understand the analysis.

B. More importantly, these are all functional measures of the length constant of calcium signals; none are estimates of either the electrical length constant of the neurites or the length constant of the synaptic output (glutamate or glycine release). The authors acknowledge the later issue to some extent, but they kind of sweep it under the rug. Yes, calcium is related to synaptic output, but this relationship can be highly local in space (more so than is captured by GCaMP6f), and it is almost always highly nonlinear, with a hill coefficient of 4 or so. Given these uncertainties, which are even more uncertain in a mixed population of glutamate and glycine synapses, I would argue that calcium signals are a very indirect measure of synaptic output. This indirectness should be reflected in the way the authors write about this relationship.

Your points are well taken. We have changed the way we report our modeling to “predicting the calcium polarity of neurites presynaptic to RGCs” as opposed to modeling the VG3-RGC drive. We have also flipped the discussion section where we mentioned uncertainty about the effect of VG3-RGC synapses to be specifically about the uncertainty in translating dendritic calcium levels into synaptic drive.

C. The question of the assumed electrical length constant vs. calcium is potentially more interesting and addressable with an additional modeling component. As the authors point out, they do not know the full complement of ion channels on VG3 neurites, so they can't make an accurate active model. However, they can make a passive cable model of the VG3 arbor. All they need for this model is the membrane resistivity and the axial resistance (which is typically assumed to be proportional to neurite diameter with a known proportionality constant). Patch-clamp recordings from previous studies should constrain the membrane resistivity based on the total resistance of a VG3 cell, but even in the absence of this measurement, the authors could test a realistic range of values based on measurements from other amacrine cells or RGCs. Since the authors have 3D reconstructions of the neurites, they are in an excellent position to build such a passive model. I suspect that the electrotonic length constant in such a model within any reasonable range of the parameter values would still be dramatically larger than the value of ~16 μm that they arrived at in the functional measurements. If I am wrong and some aspect of VG3 neuritic morphology helps to make it especially electrically isolated, that is a fascinating result that deserves more analysis. If I am right and the passive electrical properties of the VG3 cable model are incompatible with the functional data, then this itself is an important result worthy of reporting. This result would suggest that active properties of the VG3 neurites and/or a highly nonlinear relationship between voltage and calcium influx lead to greater functional isolation in the VG3 than one might expect.

We agree with the reviewer and therefore performed a significant amount of additional modeling of the electrotonic properties of VG3s. We used NEURON to model our plexus

of VG3s while varying membrane resistance, synaptic strength, and the ratio of inhibition during ON and OFF excitation. We found several conditions in which the ON/OFF ratio of maximum membrane voltage changes came close to the polarity we observed in calcium recordings. In the absence of inhibition, differences in polarity within arbors could only be achieved if the membrane conductance was unrealistically high (length constant ~ 35 μm). With inhibition, however, differences in voltage response polarity could be achieved under physiologically realistic conditions. Our results argue that most of difference in polarity observed within individual VG3s can be achieved via inhibitory shunting. We have added a results section to describe these results and panels d-f in Figure 6.

Minor suggestion

1. Line 202: *alpha character not printed correctly*

Fixed

Reviewer #2 (*Remarks to the Author*):

This study examines the link between the functional property of VG3 dendrites and their spatial organization of synaptic inputs and outputs. The authors first did subcellular two-photon calcium imaging to determine the polarity of the light responses across dendritic locations in the On and Off strata in the inner plexiform layer (IPL), and then performed connectomic reconstruction of the imaged regions. They analyzed the spatial distributions of VG3 input and output synapses across the IPL depth, and evaluated how this wiring pattern contributes to the functional properties of dendritic subregions. This is a powerful approach that fills the missing functional information in many conventional connectomic analyses. This study provides valuable information about the synaptic basis of VG3 circuit function and uncovers several intriguing features of VG3 synaptic organization, such as the distinct patterns of excitatory and inhibitory synapse distribution, biased innervation by bipolar cell type 3a, and shared bipolar cell inputs between VG3 and its postsynaptic RGC targets. It's an impressive effort and sets an example of combining functional characterization and connectomic analysis to uncover synaptic mechanisms of visual processing. I only have some minor suggestions to improve the clarity of the manuscript.

We thank the reviewer for their careful reading, positive remarks, and constructive criticism of our manuscript. We address specific comments below.

Given that $\sim 74\%$ of the synaptic inputs onto the VG3 cell are from amacrine cells, it leaves the reader wondering what these inhibitory inputs come from (i.e., from what amacrine cell types, and how they shape the dendritic responses. More information about these points will further strengthen this study.

We agree that a better understanding of the inhibitory input is critical to understanding VG3 functioning. However, the complexity of the inhibitory input requires a second study (that we are working on) and cannot be squeezed into the current publication.

That being said, we have added an inhibitory component to our modelling to show the potential for inhibitory shunting to shorten the functional length constant of VG3s.

Lines 219-220: “ we randomly pruned synapses so that synapses between pairs of connected cells were at least 10 micron from their nearest neighbor.” Can the authors provide more information about the method of random pruning and the rationale behind it? Why and how does this method “reduce biases introduced by observing only a few examples of each post synaptic cell type”?

We expanded this explanation to “A potential source of bias in comparing a depth-dependent Monte Carlo model of synapse formation to the observed data is that the formation of a real synapse between the VG3 plexus and a particular RGC arbor means that there is an increased probability of nearby opportunities for more synapses to be formed between the same pairing. To reduce this bias, we pruned observed synapses so that synapses between the same pair of cells were at least 10 mm away from each other.”

Running the model with and without the pruning gives the same results. Hopefully the explanation makes it more clear why the model with pruning is more reliable.

Lines 225-231: Given the EM dataset cannot distinguish between glutamatergic and glycinergic outputs of the VG3, the percentages don't necessarily reflect the VG3's contribution to the excitatory inputs. Along the same lines, lines 313-315, for the example of monostratified On cells, it is also possible that the VG3 synapses biased towards Off responses are inhibitory. This complication might contribute to the difference between the bipolar cell population directly connected to the RGC and the mix that is predicted to influence it through the VG3s (lines 346-348).

We agree with this point and have revised the results and discussion to reflect the ambiguity of potential synaptic drive.

Lines 351-355: I found these sentences difficult to understand.

We have rewritten this section to improve clarity:

“Given the ubiquity of the BC-VGC-RGC triad connectivity, we asked if this circuit motif reflects the kind of synapse-specific triad motif observed in the dorsal lateral geniculate nucleus³³⁻³⁵ and other retinal circuits^{36,37}. That is, are the synapses that constitute the triad clustered together within a micrometer or so of each other? We found that the connections between the bipolar cells and the VG3 and RGC were often from the same ribbon synapse. Consistent with other ribbon synapse dyads in the retina, each ribbon synapse innervating a VG3 also innervated one (222 of 259) or two (37 of 259) other neurons. The non-VG3 cell innervated by 48% of these ribbon synapses was an RGC. Of these RGC innervating ribbons, 89.5% (119 of 133) innervated RGCs that were also known to be innervated by the VG3 plexus (Fig. 7f). Thus, the observation that the same ribbon synapse innervated a VG3 and an RGC was a strong predictor that the VG3 would also innervate the RGC. However, the synapses connecting the VG3 to the

RGC did not seem especially targeted to the location of the ribbon dyad. The median distance between the bipolar-VG3 input and VG3-RGC output for all triad motifs was 13.0 μm with only 5.6% under 1 μm (Fig. 7g). Therefore, most BC-VG3-RGC triad motifs appear to operate at a local, but not synapse-specific, level of feedforward integration."

Fig. 6 "d" label is missing

Corrected

Reviewer #3 (Remarks to the Author):

The article by Friedrichsen et al entitled "Subcellular pathways through VG3 amacrine cells provide regionally tuned object-motion-sensitive signals in mouse retina" combined functional imaging and connectomic reconstruction and revealed synaptic input and output profiles in VGluT3-expressing amacrine cells (VG3s). The model evaluations of the mapped connectivity suggested a subcellular wiring diagram of input and output connectivity through a VG3 plexus.

As I read the introduction part, I understood that an appeal of this manuscript is the combination of physiological imaging and connectomics. However, individual results are not correlated well to present "novel" insight into retinal processing. Indeed, most EM analyses (Figure 2-7) were evaluated by modeling and independent of the imaging results. For example, an important correlation between physiology and anatomy could be to examine how the identified input synapses (spatial profiles, cell types) are integrated and establish the dendritic activity required for the looming sensitivity. While the authors discovered a prominent input from type 3a bipolar cells out of other cell types, how such dominated excitatory inputs can be correlated with actual VG3 responses was not examined. The authors showed an analysis to correlate characterized physiology (ON and OFF response property) and the mapped synapses (Figure 1). However, this response polarity shift is not a novel result (e.g. Chen et al., PNAS, 2017; Hsiang et al., eLife, 2017).

Also, the authors suggested a function of VG3 plexus, rather than a single cell unit, to provide local object-motion-sensitive signals with postsynaptic retinal ganglion cells (RGCs). However, the contributions of VG3 to RGC response were not substantial (Figure 6f). Indeed, previous studies have shown that only small subsets of RGC types inherit the looming sensitivity of VG3. In that sense, the impacts of the VG3 plexus would be negligible for most RGCs innervating the plexus. Further details to be considered are described below.

We thank the reviewer for their careful reading, positive remarks, and constructive criticism of our manuscript. We address specific comments below.

Major comments:

1) The correlation between physiological imaging and anatomical synapse mapping. The authors engaged in the anatomical analysis: the synaptic input mapping (Figure 3) and the spatial profiles (Figure 4). On the other hand, the relationship between the characterized synapses (locations, cell types) and physiology (dendritic activity of the individual VG3s) is unclear. Although the authors mapped the response polarity of the dendrites at the different IPL depths, this has already been shown by previous studies (e.g. Chen et al (PNAS, 2017); Hsiang et al (eLife, 2017)). To add new insights into dendritic computation in VG3, the authors should correlate those anatomical and physiological findings and propose how the mapped synaptic inputs form the dendritic activity. In particular, it is important to discuss how the identified bipolar cell (and amacrine cell) inputs are correlated with the known synaptic input properties generating looming sensitivity.

The looming responses of VG3 cells depend on fast excitation from OFF bipolar cells followed by delayed inhibition from OFF amacrine cells (Kim et al. 2020). Our present study revealed that excitatory input to VG3 dendrites is dominated by a particularly fast OFF bipolar cell type (i.e., type 3a). This choice of input partner may help open the time window between excitation and inhibition that supports strong looming responses.

Another important feature of the VG3 looming responses is that they are largely restricted to dendrites in the OFF sublamina. Our present study revealed that VG3 dendrites contain an unusually high density of inhibitory inputs (i.e., the ratio of inhibition to excitation [I/E] by synapse count is ~ 3). For other retinal neurons (incl. starburst amacrine cells) I/E ratios ~ 1 have been reported. Furthermore, we find that inhibitory synapses are enriched in the center of the VG3 dendrite arbor, between the ON and OFF sublamina. We hypothesized that the abundance and placement of inhibition could limit mixing of ON and OFF pathways, which is critical for looming selective responses and for restricting those responses to VG3 dendrites in the OFF sublamina. To test this hypothesis and explore the link between synapse patterns and function, we performed NEURON modeling in our revisions, leveraging our high-resolution reconstructions of arbor morphology and synaptic connectivity. Results from this modeling support our hypothesis. We present these results in Figure 6 in our revised manuscript.

2) The authors found that the density of amacrine cell inputs was highest between ON and OFF bipolar cell input zones, and suggested that the inhibitory inputs from amacrine cells could contribute to the ON and OFF segregation (line 115-121). If this is true, the dendritic excitation (Ca^{2+} signal) around the middle of VG3 dendritic arbors could be shunted and resulted in less Ca^{2+} signal, compared with the proximal (OFF response) and distal (ON response) dendrites. However, the response polarity mapping (Figure 1d; Chen et al., PNAS, 2017) showed ON-OFF responses at the middle VG3 dendritic arbors ("Depth $\sim 0.535, 0.536$ " in Figure 1d left). Also, previous studies (Chen et al., PNAS, 2017; Hsiang et al., eLife, 2017) did not show evident differences in Ca^{2+} signal intensity across dendritic segments. The authors should discuss how the spatial profile of amacrine cell inputs can contribute to the observed response polarities.

We thank the reviewer for raising this point. Because of the way our previous data were normalized, response amplitudes cannot be meaningfully compared across IPL depths in the figures of Hsiang et al. (2017). Importantly, in that dataset of Hsiang et al. (2017) VG3 dendrites were distinguished from background regions of the image was based on the GCaMP6f signal. Thus, non-responsive regions of the dendrites or regions in which GCaMP6f signals fall below threshold would not be included in the analysis.

To address the reviewer's comment we, therefore, performed a new set of experiments, labeling individual VG3 cells in VG3/Ai148 transgenic mice with tdTomato by in vivo electroporation. This allowed us to segment VG3 dendrites based on their tdTomato fluorescence (i.e., independent of their activity). We then analyzed how the amplitude of GCaMP6f responses to ON and OFF visual stimuli (ΔF norm.) changes across IPL depth within an individual VG3 dendrite arbor. Consistent with the distributions of excitatory and inhibitory input synapses (and the expectations of the reviewer) this revealed a dip in response amplitudes in the center of the arbor. We include our new experiments and analyses in a new supplementary figure (Supplementary Fig. 3) in our revised manuscript.

3) Based on the differences in the observed synapses and Monte Carlo (Figure 4f), the authors revealed an evident synaptic input from type 3a bipolar cells, compared with other bipolar cell types. However, it is unclear whether the probabilities of synapse formations are different across bipolar cell types or whether the density of dendritic arbors of VG3s at the different IPL depths affects the number of synapses. For example, a sparser plexus of the proximal dendrites, in which type 3a bipolar cell innervates, than the distal plexus could result in distinct synapse numbers. The authors should clarify whether the dendritic morphology affects it or not.

The Monte Carlo model we used accounts for the degree of stratification overlap between each type of bipolar cell and the VG3 arbors. We have tried to clarify this point: "We used a Monte Carlo analysis to determine how much this preference deviated from indiscriminate synapse formation. In this analysis, the observed number of synapses between bipolar cells and VG3s were randomly reassigned to bipolar cell types. The probability of synapses being assigned to a given type was determined by the degree of stratification overlap between the bipolar cell type (based on EyeWire²⁶) and our VG3 plexus (Fig. 4e)."

4) VG3s innervation to RGCs. The authors found that VG3 innervates different RGC types (Figure 5). Although 5ti and 4ow types inherit a VG3 function (sensitivity to looming stimuli), other RGC types do not. The authors should discuss why the VG3 innervations are not functional in the RGC types regardless of the unignorable VG3 inputs.

We thank the reviewer for raising this point. The VG3's sensitivity to looming stimuli is mostly restricted to dendrites in the OFF sublamina. Thus, VG3s likely send looming-sensitive signals only to RGCs with dendrites in the OFF sublamina. In Kim et al. 2020, we show that VG3 inputs shape the looming responses of 5ti (W3) and 4ow (tOFF α)

RGCs. Our present study shows that 5ti and 4ow receive abundant synaptic input from VG3s. By contrast, most other RGCs with dendrites in the OFF sublamina form few synapses with VG3s. In response to the reviewer's comments, we have rewritten our manuscript to clarify that we do not expect sparse VG3 inputs to dominate the feature selectivity of the respective targets.

Nonetheless it is possible that RGCs other than 5ti and 4ow respond to looming stimuli and that VG3s contribute to these responses. Unfortunately, the looming responses of RGCs have not been studied comprehensively and our functional connectomics dataset does not include RGC responses. We think that experimental analyses of RGC responses to looming stimuli and the VG3's contribution are beyond the scope of our present study. We hope that the reviewer is okay with this choice.

5) The authors modeled bipolar cells-RGCs transmission through VG3 (Figure 6), and claimed that there were deviations between the predicted RGC response polarity and the actual response despite the strong influence of type 3a inputs (Figure 6f). However, each RGC type receives not only bipolar inputs through VG3 (bipolar-VG3-RGC), but also direct bipolar cell inputs (bipolar-RGC). If the impact of bipolar-VG3-RGC pathways is minor compared with that of bipolar-RGC pathways, this model cannot explain the RGC response well. The authors should clarify the validity of this modeling to analyze the impact of bipolar cell inputs on RGC properties.

The predicted response properties of the VG3 to RGC synapses was not meant to predict the overall response properties of the RGCs. Rather, we were trying to figure out the extent to which the VG3 contribution to the RGC matched the overall polarity. We have tried to clarify this perspective by adding "It is expected that the majority of the RGC response polarities reflect which bipolar cell types they are directly innervated by." We have also added text discussing the potential strength or excitatory/inhibitory sign of the seemingly mismatched connections.

6) The authors claimed that a VG3 plexus provides local object-motion-sensitive signals with the postsynaptic RGC types. However, this should be examined experimentally. There is no "direct" evidence of the localization of the signal (only models in Figure 6). Even if RGCs innervating the VG3 plexus receive such local VG3 inputs, it is not evident if the VG3 inputs affect the RGC responses. Indeed, the excitatory impacts of VG3 to explain the postsynaptic RGC response were not high (Figure 6f). Therefore, to clarify the function of VG3 plexus to provide local object-motion-sensitive signals, the authors should perform Ca²⁺ imaging from the targeted RGC dendrites and examine the local dendritic activity in the looming stimuli.

In Hsiang et al. (2017), we showed that all VG3 dendrites show robust object-motion-sensitive signals. We assume that these signals are communicated through the synaptic output of VG3s (which can be excitatory or inhibitory). Particularly for RGCs receiving sparse input from VG3s, we do not assume that the selectivity of their VG3 input dominates their spike responses. As outlined in our response to the previous point, we have rewritten our manuscript to clarify our interpretation as suggested by the reviewer.

Minor comments:

1) There are no explanations of "AMC" and "BPC" (Figure 2); "VGC" (Figure 6).

Fixed

2) Figure 4. It is not clear why the population of bipolar cell types in individual VG3s looks very different from the population in "All cells". For example, the ratio of bc3a inputs is the highest in all cells. However, the populations in the individual cells are not high.

Thanks very much for the catch. The individual cell pie charts from Figure 5 (RGC) were inserted into both figure 4 and figure 5. We have corrected the figure.

3) Figure 4f. To examine the physiological relevance of the less ON bipolar cell inputs, the authors should analyze the differences in intensities of ON and OFF dendritic Ca²⁺ signals.

Most of our analysis looks at the polarity of VG3 dendrites (ON-OFF)/(ON+OFF). We have also added a supplementary figure where we show absolute calcium responses to ON and OFF stimuli (not polarity).

4) Labels of individual VG3s in Figure 5b right (e.g. cid = 2) should be related to Figure 4d right (e.g. #2).

Fixed

REVIEWERS' COMMENTS

Reviewer #1 (Remarks to the Author):

My major concerns from the previous round of review have all been addressed. I commend the authors for their thoughtful writing and the addition of the electronic modeling. The result and speculation about the importance of the inhibitory shunt in effectively isolating the VG3 neurites is very interesting.

My comments now are all minor, editorial stuff:

1. Typo on line 35 "VG3s are respond"
2. Typo on line 119 "amplitdues"
3. Line 120, "Supplemental Fig. XX" missing fig number.
4. Typo on line 244, extra ')' at the end.
5. Kind of a matter of taste, but I like calling the processes of amacrine cells "neurites" instead of "dendrites" since they contain both input and output synapses (which, of course, is a major focus of this work). But it's semantics. Perhaps we, as a field, need to change how we think about the word "dendrite".

Reviewer #2 (Remarks to the Author):

The authors have addressed my previous comments.

Reviewer #3 (Remarks to the Author):

The revision has been performed carefully with additional modeling and experiments. In particular, our major comment regarding possible physiological function suggested by the anatomical finding in this manuscript has been considered by modeling, and the authors suggest enriched inhibitory synapses in the center of VG3 dendritic arbors to segregate ON and OFF contamination. I don't have any further comments.

Thanks again for your careful reading and comments.

REVIEWERS' COMMENTS

Reviewer #1 (Remarks to the Author):

My major concerns from the previous round of review have all been addressed. I commend the authors for their thoughtful writing and the addition of the electronic modeling. The result and speculation about the importance of the inhibitory shunt in effectively isolating the VG3 neurites is very interesting.

Thank you.

My comments now are all minor, editorial stuff:

1. Typo on line 35 "VG3s are respond"

Fixed

2. Typo on line 119 "amplitdues"

Fixed

3. Line 120, "Supplemental Fig. XX" missing fig number.

Fixed

4. Typo on line 244, extra ')' at the end.

Fixed

5. Kind of a matter of taste, but I like calling the processes of amacrine cells "neurites" instead of "dendrites" since they contain both input and output synapses (which, of course, is a major focus of this work). But it's semantics. Perhaps we, as a field, need to change how we think about the word "dendrite".

Point taken. We debated using "neurites" or "dendrites" for amacrine cells. While we agree that the conventional axon/dendrite functional distinction doesn't work well for amacrine cells, the solution may be greater acceptance that the biologically important distinction between axons and dendrites is not whether they receive or transmit signals.

Reviewer #2 (Remarks to the Author):

The authors have addressed my previous comments.

Reviewer #3 (Remarks to the Author):

The revision has been performed carefully with additional modeling and experiments. In particular, our major comment regarding possible physiological function suggested by the anatomical finding in this manuscript has been considered by modeling, and the authors suggest enriched inhibitory synapses in the center of VG3 dendritic arbors to segregate ON and OFF contamination. I don't have any further comments.

Thanks.